# Rapid computations of spectrotemporal prediction error support perception of degraded speech

**Ediz Sohoglu[1]\*, Matthew H Davis[2]**

[1]School of Psychology, University of Sussex, Brighton, United Kingdom; [2]MRC Cognition and Brain Sciences Unit, Cambridge, United Kingdom

**Abstract** Human speech perception can be described as Bayesian perceptual inference but how are these Bayesian computations instantiated neurally? We used magnetoencephalographic recordings of brain responses to degraded spoken words and experimentally manipulated signal quality and prior knowledge. We first demonstrate that spectrotemporal modulations in speech are more strongly represented in neural responses than alternative speech representations (e.g. spectrogram or articulatory features). Critically, we found an interaction between speech signal quality and expectations from prior written text on the quality of neural representations; increased signal quality enhanced neural representations of speech that mismatched with prior expectations, but led to greater suppression of speech that matched prior expectations. This interaction is a unique neural signature of prediction error computations and is apparent in neural responses within 100 ms of speech input. Our findings contribute to the detailed specification of a computational model of speech perception based on predictive coding frameworks.

## Introduction

Although we understand spoken language rapidly and automatically, speech is an inherently ambiguous acoustic signal, compatible with multiple interpretations. Such ambiguities are evident even for clearly spoken speech: A /t/ consonant will sometimes be confused with /p/, as these are both unvoiced stops with similar acoustic characteristics (*Warner et al., 2014*). In real-world environments, where the acoustic signal is degraded or heard in the presence of noise or competing speakers, additional uncertainty arises and speech comprehension is further challenged (*Mattys et al., 2012*; *Peelle, 2018*).

Given the uncertainty of the speech signal, listeners must exploit prior knowledge or expectations to constrain perception. For example, ambiguities in perceiving individual speech sounds are more readily resolved if those sounds are heard in the context of a word (*Ganong, 1980*; *Rogers and Davis, 2017*). Following probability theory, the optimal strategy for combining prior knowledge with sensory signals is by applying Bayes theorem to compute the posterior probabilities of different interpretations of the input. Indeed, it has been suggested that spoken word recognition is fundamentally a process of Bayesian inference (*Norris and McQueen, 2008*). This work aims to establish how these Bayesian computations might be instantiated neurally.

There are at least two representational schemes by which the brain could implement Bayesian inference (depicted in *Figure 1C*; see *Aitchison and Lengyel, 2017*). One possibility is that neural representations of sensory signals are enhanced or 'sharpened' by prior knowledge (*Murray et al., 2004*; *Friston, 2005*; *Blank and Davis, 2016*; *de Lange et al., 2018*). Under a sharpening scheme, neural responses directly encode posterior probabilities and hence representations of speech sounds are enhanced in the same way as perceptual outcomes are enhanced by prior knowledge (*McClelland and Elman, 1986*; *McClelland et al., 2014*). Alternatively, neural representations of

**\*For correspondence:**
E.Sohoglu@sussex.ac.uk

**Competing interests:** The authors declare that no competing interests exist.

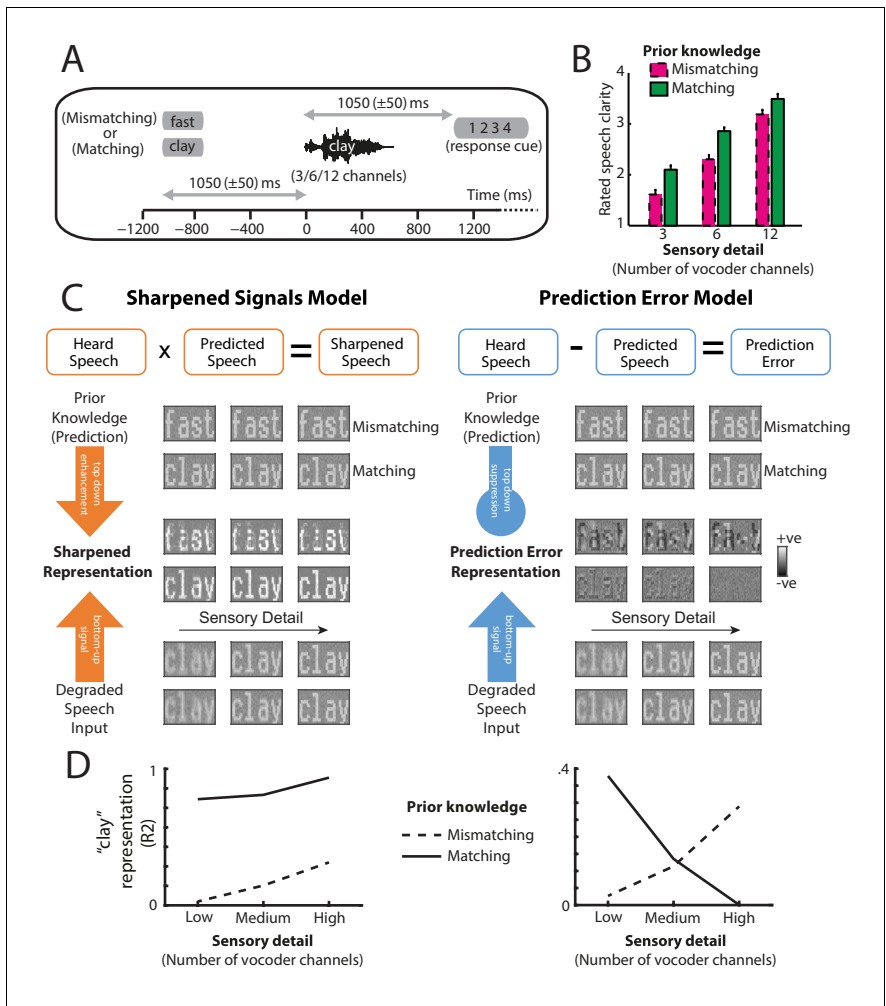

**Figure 1.** Overview of experimental design and hypotheses. (A) On each trial, listeners heard and judged the clarity of a degraded spoken word. Listeners' prior knowledge of speech content was manipulated by presenting matching ('clay') or mismatching ('fast') text before spoken words presented with varying levels of sensory detail (3/6/12-channel vocoded), panel reproduced from B, *Sohoglu and Davis, 2016*. (B) Ratings of speech clarity were enhanced not only by increasing sensory detail but also by prior knowledge from matching text (graph reproduced from Figure 2A, *Sohoglu and Davis, 2016*). Error bars represent the standard error of the mean after removing between-subject variance, suitable for repeated-measures comparisons (*Loftus and Masson, 1994*). (C) Schematic illustrations of two representational schemes by which prior knowledge and speech input are combined (for details, see Materials and methods section). For illustrative purposes, we depict degraded speech as visually degraded text. Under a sharpening scheme (left panels), neural representations of degraded sensory signals (bottom) are enhanced by matching prior knowledge (top) in the same way as perceptual outcomes are enhanced by prior knowledge. Under a prediction error scheme (right panels), neural representations of expected speech sounds are subtracted from sensory signals. These two schemes make different predictions for experiments that assess the content of neural representations when sensory detail and prior knowledge of speech are manipulated. (D) Theoretical predictions for sharpened signal (left) and prediction error (right) models. In a sharpened signal model, representations of the heard spoken word (e.g. 'clay'; expressed as the squared correlation with a clear [noise-free] 'clay') are most accurately encoded in neural responses when increasing speech sensory detail and matching prior knowledge combine to enhance perception. Conversely, for models that represent prediction error, an interaction between sensory detail and prior knowledge is observed. For speech that mismatches with prior knowledge, increasing sensory detail results in better representation of the heard word 'clay' because bottom-up input remains unexplained. Conversely, for speech that matches prior knowledge, increased sensory detail results in worse encoding of 'clay' because bottom-up input is explained away. Note that while the overall magnitude of prediction error is always the smallest when expectations match with speech input (see *Figure 1— figure supplement 1*), the prediction error representation of matching 'clay' is enhanced for low-clarity speech and diminished for high-clarity speech. For explanation, see the Discussion section.

*Figure 1 continued on next page*

*Figure 1 continued*

The online version of this article includes the following figure supplement(s) for figure 1:

**Figure supplement 1.** Summed absolute prediction error for representations illustrated in *Figure 1C*.

expected speech sounds are subtracted from bottom-up signals, such that only the unexpected parts (i.e. 'prediction error') are passed up the cortical hierarchy to update higher-level representations (*Rao and Ballard, 1999*). Under this latter representational scheme, higher-level neural representations come to encode posterior probabilities (as required by Bayesian inference) but this is achieved by an intermediate process in which prediction errors are computed (*Aitchison and Lengyel, 2017*). In many models, both representational schemes are utilized, in separate neural populations (cf. predictive coding, *Rao and Ballard, 1999*; *Spratling, 2008*; *Bastos et al., 2012*).

A range of experimental evidence has been used to distinguish sharpened and prediction error representations. An often observed finding is that matching prior knowledge reduces the amplitude of evoked neural responses (*Ulanovsky et al., 2003*; *Grill-Spector et al., 2006*; *Rabovsky et al., 2018*). This observation is commonly attributed to prediction errors since expected stimuli are relatively predictable and therefore should evoke reduced prediction error. However, reduced activity is equally compatible with sharpened responses because, under this representational scheme, neuronal activity encoding competing features (i.e. 'noise') is suppressed (see *Figure 1C*; *Murray et al., 2004*; *Friston, 2005*; *Blank and Davis, 2016*; *Aitchison and Lengyel, 2017*; *de Lange et al., 2018*).

One way to adjudicate between representations is by manipulating signal quality alongside prior knowledge and measuring the consequences for the pattern (rather than only the mean) of neural responses. Computational simulations reported by *Blank and Davis, 2016* demonstrate a unique hallmark of prediction errors which is that neural representations of sensory stimuli show an interaction between signal quality and prior knowledge (see *Figure 1D*). This interaction arises because sensory signals that match strong prior expectations are explained away more effectively as signal quality increases and hence neural representations are suppressed even as perceptual outcomes improve. Whereas for sensory signals that follow uninformative prior expectations, increased signal quality leads to a corresponding increase in sensory information that remains unexplained (see *Figure 1C*). This pattern – opposite effects of signal quality on neural representations depending on whether prior knowledge is informative or uninformative – highlights an important implication of neural activity that represents prediction errors. Rather than directly signaling perceptual outcomes, these neural signals in the sensory cortex serve the intermediary function of updating higher-level (phonological, lexical, or semantic) representations to generate a perceptual interpretation (the posterior, in Bayesian terms) from a prediction (prior). It is these updated perceptual interpretations, and not prediction errors, that should correlate most closely with perceived clarity (*Sohoglu and Davis, 2016*).

By contrast, in computational simulations implementing a sharpening scheme, representational patterns are similarly enhanced by increased signal quality and matching prior knowledge (see *Figure 1D*; *Murray et al., 2004*; *Friston, 2005*; *Blank and Davis, 2016*; *Aitchison and Lengyel, 2017*; *de Lange et al., 2018*). Using prior written text to manipulate listeners' prior knowledge of degraded spoken words, *Blank and Davis, 2016* showed that multivoxel representations of speech in the superior temporal gyrus (as measured by fMRI) showed an interaction between signal quality and prior expectations, consistent with prediction error computations. This is despite the observation that the mean multivoxel responses to speech were always reduced by matching expectations, no matter the level of sensory detail.

Although the study of *Blank and Davis, 2016* provides evidence in support of prediction errors, key questions remain. First, it remains unclear at which levels of representation prediction errors are computed. Predictive coding models that utilize prediction errors are hierarchically organized such that predictions are signaled by top-down connections and prediction errors by bottom-up

connections. Therefore, in principle, prediction errors will be computed at multiple levels of representation. Previous studies using similar paradigms have suggested either a higher-level phonetic (*Di Liberto et al., 2018a*) or a lower-level acoustic locus of prediction error representations (*Holdgraf et al., 2016*). Second, due to the sluggishness of the BOLD signal, the timecourse over which prediction errors are computed is unknown. Therefore, it is unclear whether prediction errors are formed only at late latencies following re-entrant feedback or more rapidly during the initial feedforward sweep of cortical processing (*Sohoglu and Davis, 2016*; *Kok et al., 2017*; *de Lange et al., 2018*; *Di Liberto et al., 2018a*).

In this study, we reanalyzed MEG recordings of neural activity from a previous experiment in which we simultaneously manipulated signal quality and listeners' prior knowledge during speech perception (*Sohoglu and Davis, 2016*). Listeners heard degraded (noise-vocoded) spoken words with varying amounts of sensory detail and hence at different levels of signal quality. Before each spoken word, listeners' read matching or mismatching text and therefore had accurate or inaccurate prior knowledge of upcoming speech content (*Figure 1A*). Our previously reported analyses focused on the mean amplitude of evoked responses, which as explained above, cannot adjudicate between sharpened and prediction error representations. We, therefore, used linear regression to test which of several candidate speech features are encoded in MEG responses (*Ding and Simon, 2012*; *Pasley et al., 2012*; *Crosse et al., 2016*; *Holdgraf et al., 2017*) and further asked how those feature representations are modulated by signal quality and prior knowledge. Following *Blank and Davis, 2016*, the two-way interaction between sensory detail and prior knowledge is diagnostic of prediction errors (see *Figure 1C and D*). Because of the temporal resolution of MEG, if we observe such an interaction we can also determine the latency at which it occurs.

## Results

### Behavior

During the MEG recordings, listeners completed a clarity rating task in which they judged the subjective clarity of each degraded (noise-vocoded) spoken word (*Figure 1A and B*). Ratings of speech clarity were enhanced both when sensory detail increased ($F$ (2,40) = 295, $\mu_p^2$ = .937, $p$ <.001) and when listeners had prior knowledge from matching written text ($F$ (1,20) = 93.2, $\mu_p^2$ = .823, $p$ <.001). These behavioral results have previously been reported (*Sohoglu and Davis, 2016*) but we include them here to facilitate interpretation of the present MEG analyses.

### Encoding analysis: Stimulus feature space selection

We used ridge regression to predict the MEG data from the stimulus features. A model that accurately predicts the MEG data would indicate that the component features in the model are well represented in neural responses. Four feature spaces were obtained from the original clear versions of the spoken stimuli (i.e. before noise-vocoding; see *Figure 2* and Materials and methods for a full description). Constructing the feature spaces from clear speech enabled us to more easily compare model accuracies for speech stimuli presented under different degradation levels: any model accuracy differences as a function of vocoder channels should be attributed to neural encoding rather than inherent acoustic differences in the stimulus. The first two feature spaces (envelope and spectrogram) were simple acoustic representations that captured time-varying sound energy. These were followed by more complex feature spaces with greater abstraction from the sound waveform: one acoustic (spectrotemporal modulations) and one linguistic (phonetic features). In the first stage of our analysis, we examined which of these four feature spaces best predicted the MEG data.

We averaged model accuracies over conditions and over the 20 sensors with the highest model accuracies (computed separately for each feature space, hemisphere, and participant; shown as black bars in *Figure 3A*). There was a significant main effect of feature space on model accuracies ($F$ (3,60) = 51.5, $\mu_p^2$ = .720, $p$ <.001), which did not interact with hemisphere ($F$ (3,60) = 1.56, $\mu_p^2$ = .072, $p$ = .223). Post-hoc $t$-tests revealed that the spectrotemporal modulation feature space best predicted the MEG data ($p$-values shown in *Figure 3A*). Sensor selections (i.e. those sensors with the highest model accuracies; shown in *Figure 3B*) were similarly distributed over temporal and frontal

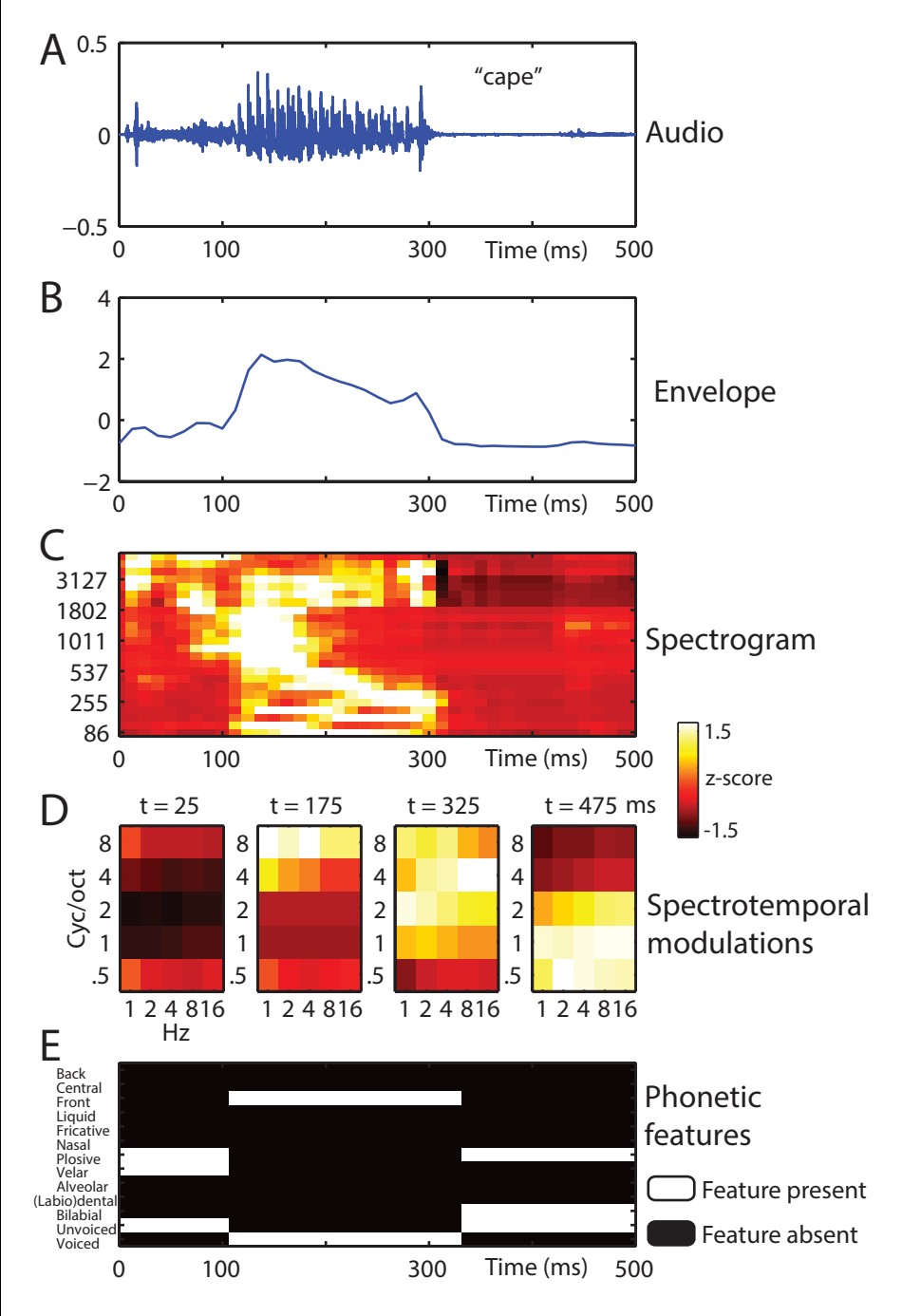

**Figure 2.** Stimulus feature spaces used to model MEG responses, for the example word 'cape'. (**A**) shown as an audio waveform for the original clear recording (i.e. before vocoding). (**B**) Envelope: broadband envelope derived from summing the envelopes across all spectral channels of a 24-channel noise-vocoder. (**C**) Spectrogram: derived from the envelope in each spectral channel of a 24-channel noise-vocoder. (**D**) Spectrotemporal modulations: derived from the spectral and temporal decomposition of a spectrogram into 25 spectrotemporal modulation channels, illustrated at regular temporal intervals through the spoken word. (**E**) Phonetic features: derived from segment-based representations of spoken words.

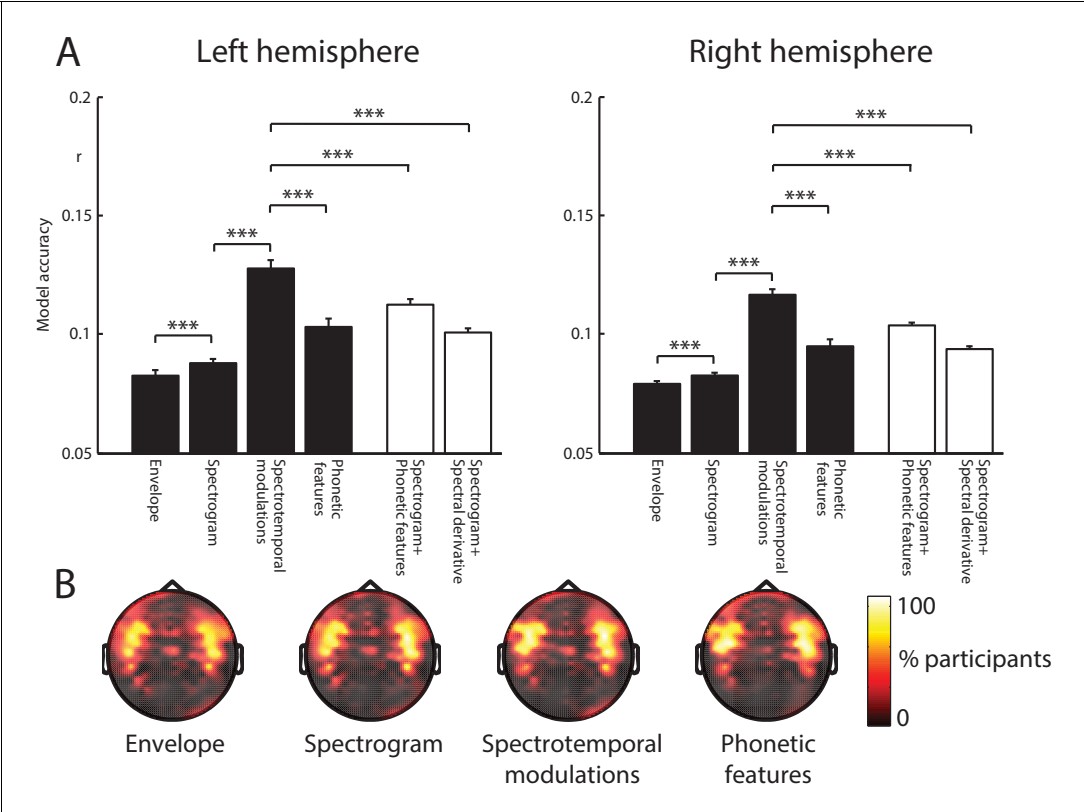

**Figure 3.** Encoding model comparison. (**A**) Mean model accuracies for the four feature spaces (black bars) and the two feature space combinations (white bars). Error bars represent the standard error of the mean after removing between-subject variance, suitable for repeated-measures comparisons (**Loftus and Masson, 1994**). Braces indicate the significance of paired *t*-tests ***p<0.001 (**B**) Topographic distribution of MEG gradiometer sensors over which model accuracies were averaged. In each hemisphere of each participant, we selected 20 sensors with the highest model accuracies. The topographies show the percentage of participants for which each sensor was selected, consistent with neural sources in the superior temporal cortex. The online version of this article includes the following figure supplement(s) for figure 3:

**Figure supplement 1.** Increases in model accuracy when combining feature spaces.

sites for all feature spaces, consistent with bilateral neural generators in the superior temporal cortex.

We additionally tested more complex combinations of feature spaces that previous work (**Daube et al., 2019**; **Di Liberto et al., 2015**) has shown to be good models of neural responses (shown as white bars in **Figure 3A**). We were particularly interested in how the model performance of the spectrotemporal modulation feature space compared with these more complex feature spaces. Despite having lower dimensionality (25 features), the spectrotemporal modulation feature space remained the model that best predicted neural responses, outperforming Spectrogram+Phonetic features (24+13 = 37 features; $F(1,20) = 16.5$, $\mu_p^2 = .453$, $p = .001$) and Spectrogram+Spectral derivative (24+24 = 48 features; $F(1,20) = 98.2$, $\mu_p^2 = .831$, $p < .001$). No effects involving hemisphere were significant (all *p*'s >.125).

Note that because different feature spaces tend to be correlated with each other, combining different feature spaces does not result in additive increases in model accuracy (**Norman-Haignere and McDermott, 2018**; **Kriegeskorte and Douglas, 2019**). For example, the model performance for Spectrogram+Phonetic features does not equal the sum of the model accuracies for the individual Spectrogram and Phonetic features models (**Figure 3A**). To more clearly identify the unique contributions of the feature spaces, we tested additional combinations of feature spaces and conducted more systematic comparisons.

We first sought to replicate previous findings by testing whether the addition of Phonetic features or the Spectral derivative could improve model performance beyond the Spectrogram alone

(*Di Liberto et al., 2015*; *Daube et al., 2019*). Results and *p*-values are shown in *Figure 3—figure supplement 1A*, expressed as performance increases over the spectrogram alone. We replicated previous findings in that both the Spectrogram+Phonetic features model and the Spectrogram +Spectral derivative model outperformed the Spectrogram feature space. We also tested whether adding the spectrotemporal modulation model to the spectrogram feature space could improve performance. This resulted in the biggest performance gain, consistent with our earlier findings indicating that spectrotemporal modulations was the feature space that individually predicted MEG responses most accurately.

Next, we tested whether combining the spectrotemporal modulation model with other feature spaces could improve model accuracy (shown in *Figure 3—figure supplement 1B*). We found that phonetic features, the spectral derivative and spectrogram all resulted in performance gains beyond spectrotemporal modulations alone. Of these three feature spaces, adding phonetic features resulted in the largest increase in model accuracy. This indicates that although spectrotemporal modulations can explain the largest MEG variance of any feature model individually, other feature spaces (phonetic features in particular) are also represented in MEG responses.

Because the performance of a spectrogram feature space can be improved by including a compressive non-linearity (*Daube et al., 2019*), as found in the auditory periphery, we also repeated the above analysis after raising the spectrogram values to the power of 0.3. While this change increased model accuracies for the spectrogram and spectral derivative feature spaces, the overall pattern of results remained the same.

## Acoustic analysis: Stimulus modulation content and effect of vocoding

Our analysis above suggests that a feature space comprised of spectrotemporal modulations is most accurately represented in neural responses. One of the motivations for testing this feature space stems from the noise-vocoding procedure used to degrade our speech stimuli, which removes narrowband spectral modulations while leaving slow temporal modulations intact (*Shannon et al., 1995*; *Roberts et al., 2011*). To investigate the acoustic impact of noise-vocoding on our stimuli, we next characterized the spectrotemporal modulations that convey speech content in our stimulus set and how those modulations are affected by noise-vocoding with a wide range of spectral channels, from 1 to 24 channels. As shown in *Figure 4A*, modulations showed a lowpass profile and were strongest in magnitude for low spectrotemporal frequencies. This is consistent with previous work (*Voss and Clarke, 1975*; *Singh and Theunissen, 2003*) demonstrating that modulation power of natural sounds decays with increasing frequency following a 1/f relationship (where f is the frequency). Within this lowpass region, different spectrotemporal modulations relate to distinct types of speech sound (*Elliott and Theunissen, 2009*). For example, fast temporal and broad spectral modulations reflect transient sounds such as stop consonant release bursts whereas slow temporal and narrowband spectral modulations reflect sounds with a sustained spectral structure such as vowels. More intermediate modulations correspond to formant transitions that cue consonant place and manner of articulation (*Liberman et al., 1967*).

Over items, increasing the number of vocoder channels resulted in significantly higher signal magnitude specific to intermediate spectral and temporal modulations (1–2 cycles per octave and 2–4 Hz, all effects shown are FDR corrected for multiple comparisons across spectrotemporal modulations; see *Figure 4C*). Thus, while low-frequency modulations dominate the speech signal overall (irrespective of the number of vocoder channels), it is the intermediate spectrotemporal modulations that are most strongly affected by noise-vocoding. These intermediate spectrotemporal modulations are known to support speech intelligibility (*Elliott and Theunissen, 2009*; *Venezia et al., 2016*; *Flinker et al., 2019*), consistent with the strong impact of the number of vocoder channels on word report accuracy (e.g. *Shannon et al., 1995*; *Davis and Johnsrude, 2003*; *Scott et al., 2006*; *Obleser et al., 2008*) and subjective clarity (e.g. *Obleser et al., 2008*; *Sohoglu et al., 2014*). The opposite effect (i.e. *decreased* signal magnitude with an increasing number of vocoder channels) was observed for broadband spectral modulations (0.5 cycles/octave) across all temporal modulation rates (reflecting the increase in envelope co-modulation when fewer spectral channels are available) and for fast temporal modulations (16 Hz) and narrowband (>2 cycles/octave) spectral modulations (reflecting stochastic fluctuations in the noise carrier; *Stone et al., 2008*).

We also asked which spectrotemporal modulations were most informative for discriminating between different words. For each spectrotemporal modulation bin, we computed the Euclidean

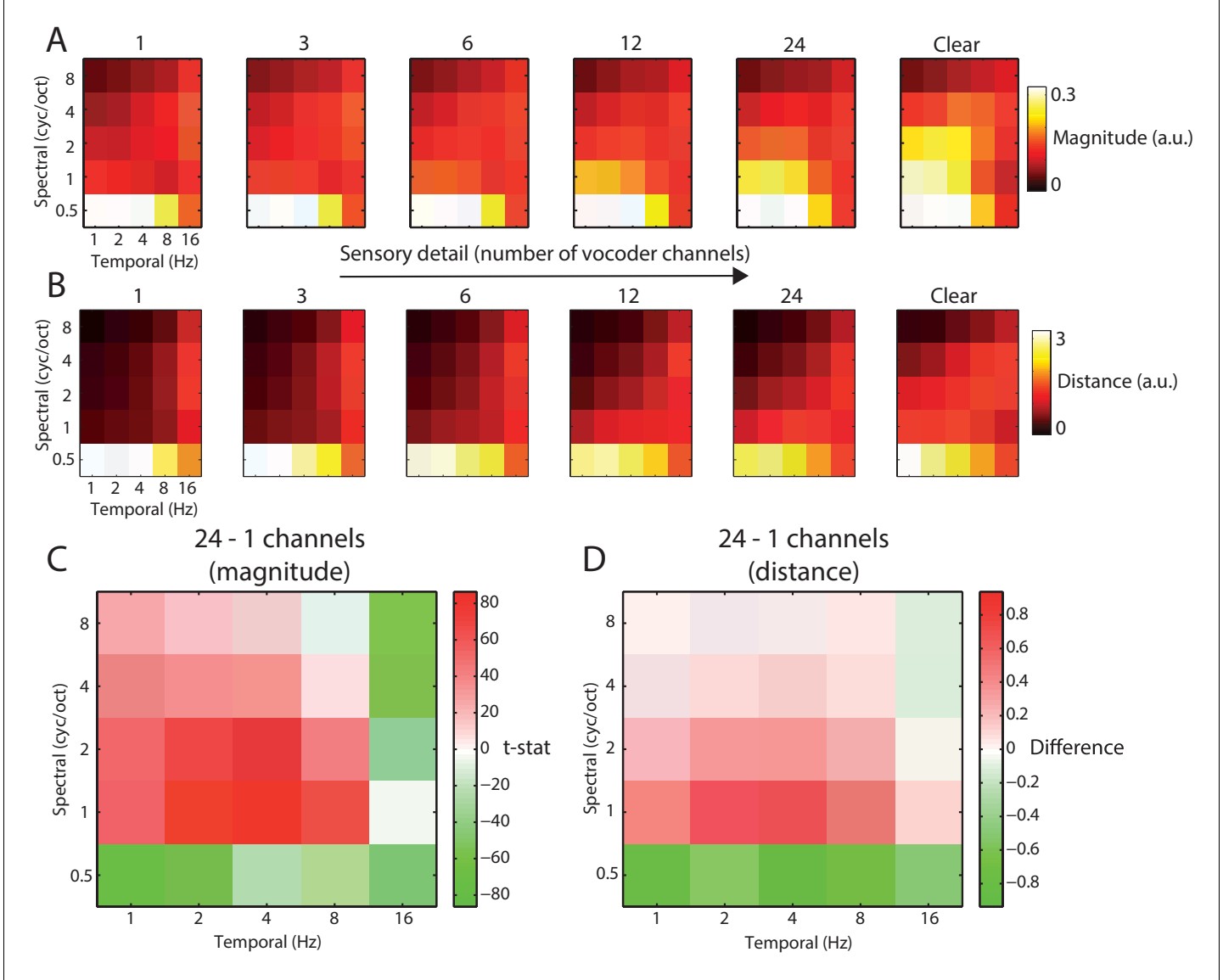

**Figure 4.** Acoustic comparison of vocoded speech stimuli at varying levels of sensory detail. (**A**) The magnitude of different spectrotemporal modulations for speech vocoded with different numbers of channels and clear speech (for comparison only; clear speech was not presented to listeners in the experiment). (**B**) Mean between-word Euclidean distance for different spectrotemporal modulations in speech vocoded with different numbers of channels. (**C**) Paired *t*-test showing significant differences in spectrotemporal modulation magnitude for comparison of 958 spoken words vocoded with 24 channels versus one channel (p<0.05 FDR corrected for multiple comparisons across spectrotemporal modulations). (**D**) Mean difference of between-word Euclidean distances for different spectrotemporal modulations for 24 versus 1 channel vocoded speech.

distances between the time series of all the words in our stimulus set and averaged the resulting distances (see *Figure 4B*). Between-word distances resembled the magnitude-based analysis, with the greatest distance for low spectrotemporal frequencies. Between-word distances also showed a similar effect of increasing the number of vocoder channels (see *Figure 4D*).

## Encoding analysis: Condition effects

Next, we determined between-condition differences in the ability of the spectrotemporal modulation feature space to predict MEG responses. Any such differences would indicate that the neural representation of spectrotemporal modulations is modulated by prior knowledge or speech sensory detail. Model accuracies for different conditions are shown in *Figure 5A*. There was a significant interaction between prior knowledge and speech sensory detail ($F(2,40) = 5.41$, $\mu_p^2 = .213$, $p = .01$),

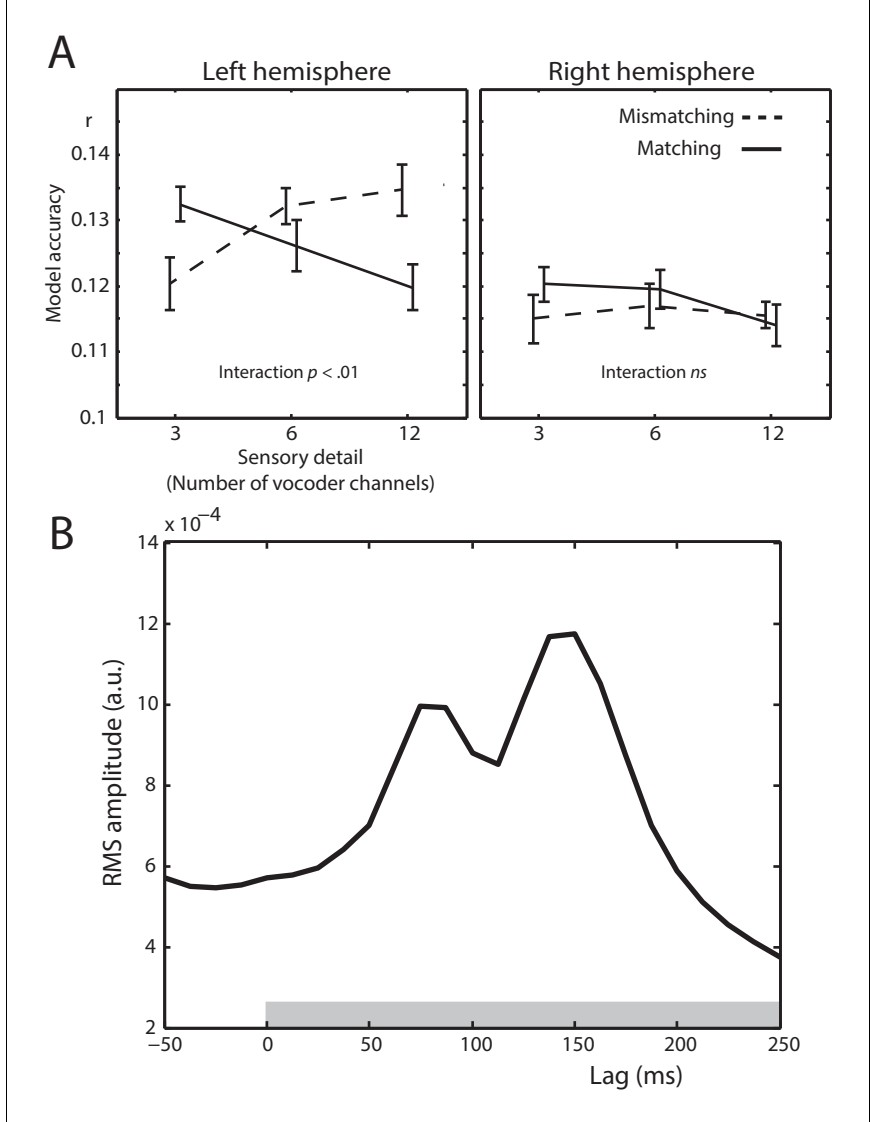

**Figure 5.** Spectrotemporal modulation encoding model results. (**A**) Mean model accuracies for lags between 0 and 250 ms as a function of sensory detail (3/6/12 channel vocoded words), and prior knowledge (speech after mismatching/matching text) in the left and right hemisphere sensors. Error bars represent the standard error of the mean after removing between-subject variance, suitable for repeated-measures comparisons (**Loftus and Masson, 1994**). (**B**) Root Mean Square (RMS) amplitude across all left hemisphere sensors for the Temporal Response Functions (TRFs) averaged over spectrotemporal modulations, conditions, and participants. The gray box at the bottom of the graph indicates the lags used in computing the model accuracy data in panel A. The online version of this article includes the following figure supplement(s) for figure 5:

**Figure supplement 1.** Control analysis of encoding model accuracies.

which marginally interacted with hemisphere ($F$ (2,40) = 3.57, $\mu_p^2$ = .151, $p$ = .051). Follow-up tests in the left hemisphere sensors again showed a statistical interaction between sensory detail and prior knowledge ($F$ (2,40) = 7.66, $\mu_p^2$ = .277, $p$ = .002). For speech that Mismatched with prior knowledge, model accuracies increased with increasing speech sensory detail ($F$ (2,40) = 4.49, $\mu_p^2$ = .183, $p$ = .021). In the Matching condition, however, model accuracies decreased with increasing sensory detail ($F$ (2,40) = 3.70, $\mu_p^2$ = .156, $p$ = .037). The statistical interaction between sensory detail and prior knowledge can also be characterized by comparing model accuracies for Matching versus

Mismatching conditions at each level of sensory detail. While model accuracies were greater for Matching versus Mismatching conditions for low clarity 3 channel speech ($t$ (20) = 2.201, $d_z$ = .480, $p$ = .040), the opposite was true for high clarity 12 channel speech ($t$ (20) = 2.384, $d_z$ = .520, $p$ = .027). Model accuracies did not significantly differ between Matching and Mismatching conditions for 6 channel speech with intermediate clarity ($t$ (20) = 1.085, $d_z$ = .237, $p$ = .291). This interaction is consistent with a prediction error scheme and inconsistent with sharpened representations (compare *Figure 5A* with *Figure 1D*). No significant differences were observed in the right hemisphere.

We also conducted a control analysis in which we compared the observed model accuracies to empirical null distributions. This allowed us to test whether neural responses encode spectrotemporal modulations even in the Mismatching three channel and Matching 12 channel conditions, that is, when model accuracies were lowest. The null distributions were created by fully permuted each spoken word's feature representation (spectrotemporal modulation channels and time-bins) and repeating this for each condition and 100 permutations. As can be seen in *Figure 5—figure supplement 1A* (only the left hemisphere sensors shown), this results in near-zero null distributions. Accordingly, observed model accuracies in all six conditions were significantly above chance as defined by the null distributions (all $p$'s < 0.01). Hence, even in the Mismatching three channel and Matching 12 channel conditions, neural responses encode spectrotemporal modulations in speech.

To test whether the interaction between prior knowledge and sensory detail in the observed data remains after accounting for the empirical null distributions, we z-scored the observed data with respect to the feature-shuffled distributions (shown in *Figure 5—figure supplement 1B*). As expected, given the near-zero model accuracies in the null distributions, the z-scored data show essentially the same pattern of results as seen previously in *Figure 5A* (prior knowledge by sensory detail interaction in the left hemisphere: $F$ (2,40) = 5.92, $\mu_p^2$ = .228, $p$ = .006).

In a further control analysis, we created additional null distributions by randomly permuting the feature representations across trials (i.e. shuffling the words in our stimulus set while keeping the component features of the words intact). The resulting null distributions for the left hemisphere sensors are shown in *Figure 5—figure supplement 1C*. Once again, observed model accuracies in all six conditions (shown as broken lines in the figure) were significantly above chance as defined by the null distributions (all $p$'s < 0.01). This confirms that neural responses encode the *specific* acoustic form of the heard spoken word. To our surprise, however, a prior knowledge by sensory detail interaction is also apparent in the null distributions (see *Figure 5—figure supplement 1C*). This suggests that a spectrotemporal representation of a generic or randomly chosen word can also predict MEG responses with some degree of accuracy. This is possible because the words used in the experiment have homogeneous acoustic properties, for example, they are all monosyllabic words spoken by the same individual. All words therefore share, to some extent, a common spectrotemporal profile.

As before, we z-scored the observed data with respect to the empirical null distributions (now from word shuffling). As shown in *Figure 5—figure supplement 1D*, an interaction is still apparent in the z-scored data. Confirming this, repeated measures ANOVA showed a significant interaction between prior knowledge and sensory detail ($F$ (2,40) = 3.41, $\mu_p^2$ = .146, $p$ = .048). Although unlike the original interaction in *Figure 5A*, this interaction was equally strong in both hemispheres ($F$ (2,40) = .324, $\mu_p^2$ = .016, $p$ = .683). In addition, simple effects of speech sensory detail were not significant (Mismatching: $F$ (2,40) = 1.67, $\mu_p^2$ = .077, $p$ = .202; Matching: $F$ (2,40) = 1.69, $\mu_p^2$ = .078, $p$ = .199) although there were marginal changes from 3 to 6 channels for both Mismatching ($F$ (1,20) = 3.33, $\mu_p^2$ = .143, $p$ = .083) and Matching ($F$ (1,20) = 3.32, $\mu_p^2$ = .142, $p$ = .083) speech.

Taken together, the above control analyses confirm that encoding of an acoustic representation of a heard word – either for the specific word spoken, or from generic acoustic elements shared with other monosyllabic words from the same speaker – shows an interaction between prior knowledge and sensory detail that is more consistent with a prediction error scheme. We will return to this point in the Discussion.

Our encoding analysis integrates past information in the stimulus (i.e. over multiple lags from 0 to 250 ms) to predict the neural response. Thus, the analysis of model accuracies above does not indicate when encoding occurs within this 250 ms period. One way to determine this is to examine the weights of the encoding model linking stimulus variation to neural responses, that is, temporal response functions (TRFs). The TRFs show two peaks at 87.5 and 150 ms (shown in *Figure 5B*). This indicates that variations in spectrotemporal modulations are linked to the largest changes in MEG

responses 87.5 and 150 ms later. These findings are consistent with previous work showing multiple early components in response to ongoing acoustic features in speech that resemble the prominent P1, N1 and P2 components seen when timelocking to speech onset (e.g. *Lalor and Foxe, 2010*; *Ding and Simon, 2012*; *Di Liberto et al., 2015*). Thus, analysis of the TRFs confirms that encoding of spectrotemporal modulations is associated with short-latency neural responses at a relatively early hierarchical stage of speech processing. In a later section (see 'Decoding analysis' below), we address when condition differences emerge.

## Decoding analysis

To link MEG responses with specific spectrotemporal modulations, we used linear regression for data prediction in the opposite direction: from MEG responses to speech spectrotemporal modulations (i.e. decoding analysis). As shown in *Figure 6A*, intermediate temporal modulations (2–4 Hz) were best decoded from MEG responses. This observation is consistent with previous neurophysiological (*Ahissar et al., 2001*; *Luo and Poeppel, 2007*; *Peelle et al., 2013*; *Ding and Simon, 2014*; *Di Liberto et al., 2015*; *Park et al., 2015*; *Obleser and Kayser, 2019*) and fMRI (*Santoro et al., 2017*) data showing that intermediate temporal modulations are well-represented in auditory cortex. These intermediate temporal modulations are also most impacted by noise-vocoding (see *Figure 4C and D*) and support speech intelligibility (*Elliott and Theunissen, 2009*; *Venezia et al., 2016*).

Having previously identified a significant interaction between prior knowledge and sensory detail in our encoding analysis, we next conducted a targeted *t*-test on decoding accuracies using the interaction contrast: 12–3 channels (Mismatch) – 12–3 channels (Match). This allowed us to identify *which* spectrotemporal modulations show evidence of the interaction between prior knowledge and sensory detail. As shown in *Figure 6B*, this interaction was observed at intermediate (2–4 Hz) temporal modulations (FDR corrected across spectrotemporal modulations). Visualization of the difference between 12 and 3 channels for Mismatch and Match conditions separately confirmed that this interaction was of the same form observed previously (i.e. increasing sensory detail led to opposite effects on decoding accuracy when prior knowledge mismatched versus matched with speech). Thus, intermediate temporal modulations are well represented in MEG responses and it is these representations that are affected by our manipulations of sensory and prior knowledge.

Our analysis up to this point does not reveal the timecourse of the neural interaction between sensory detail and prior knowledge. While the weights of the spectrotemporal encoding model implicate early neural responses (shown in *Figure 5B*), this evidence is indirect as the weights were averaged over conditions. To examine the timecourse of the critical interaction, we conducted a single-lag decoding analysis (*O'Sullivan et al., 2015*) in which we decoded spectrotemporal modulations from neural responses at each lag separately from −50 to 250 ms relative to speech input. For this analysis, we first averaged over the decoding accuracies in *Figure 6B* that previously showed the interaction when integrating over lags. As shown in *Figure 6D*, the interaction contrast 12–3 channels (Mismatch) – 12–3 channels (Match) emerged rapidly, peaking at around 50 ms. The contrast of 12–3 channels separately for Mismatch and Match conditions again confirmed the form of the interaction (i.e. opposite effects of sensory detail on decoding accuracy in Mismatch and Match trials) although these effects were only significant at an uncorrected p<0.05 level. We also directly compared Match and Mismatch trials by computing the contrast Match – Mismatch for 3 channels and 12 channels separately (shown in *Figure 6—figure supplement 1*). While we did not observe significant differences for three channel speech (even at an uncorrected p<0.05 level), for 12 channel speech decoding accuracy was lower in Match versus Mismatch trials from 25 to 40 ms and then again from 125 to 150 and 175–200 ms (albeit at uncorrected p<0.05 level).

## Discussion

Here using linear regression and MEG responses to noise-vocoded words, we report four main findings. First, spectrotemporal modulation content in the acoustic signal is well represented in MEG responses, more so than alternative speech features (envelope, spectrogram, and phonetic features). Second, the information content of speech is well characterized by spectrotemporal modulations. Modulations at intermediate temporal (2–4 Hz) and spectral scales (1–2 cycles/octave) are critical for distinguishing between individual spoken words and these spectrotemporal modulations are affected by noise-vocoding manipulations that impact speech intelligibility. Third, signal quality and

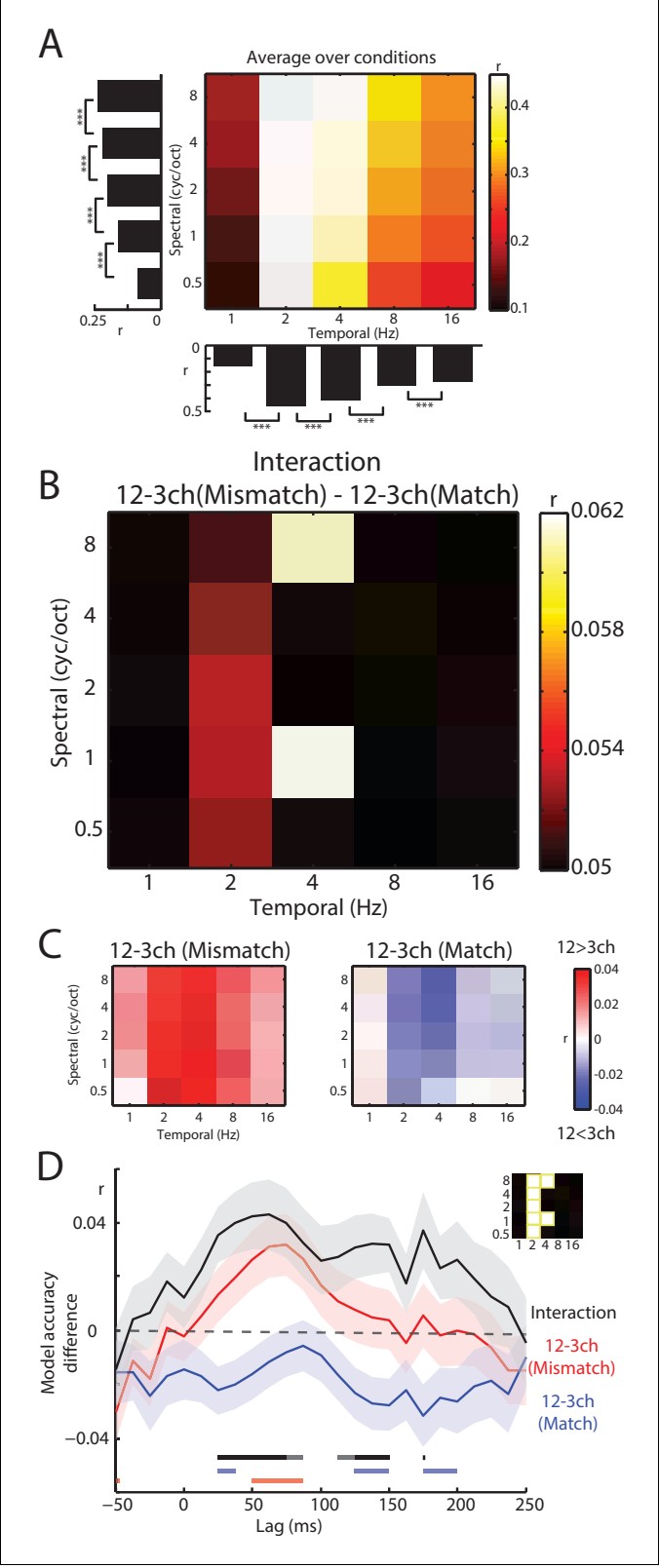

**Figure 6.** Decoding of spectrotemporal modulations from MEG responses to speech. (A) The grid shows model accuracies for specific spectrotemporal modulations averaged over conditions. The left bar graph depicts model accuracy for each spectral modulation frequency, averaged over temporal modulations. Bottom bar graph depicts model accuracy for each temporal modulation frequency, averaged over spectral modulations. Braces indicate

*Figure 6 continued on next page*

*Figure 6 continued*

the significance of paired *t*-tests ***p<0.001 (**B**) Effect size (model accuracy differences, r) for the interaction contrast: 12–3 channels (Mismatch) – 12–3 channels (Match). The effect size display has been thresholded so as to only show cells in which the sensory detail by prior knowledge interaction is statistically significant at p<0.05 FDR corrected for multiple comparisons across spectrotemporal modulations. (**C**) Effect size (model accuracy differences, r) for comparisons between 12 and 3 channels, computed separately for Mismatch and Match conditions. Red shows greater model accuracy for 12 channel than for three channel speech (observed for speech that Mismatches with written text). Blue shows lower model accuracy for 12 channel than for three channel speech (observed for speech that matches written text). ch = channels. (**D**) Timecourse of decoding accuracy (single-lag analysis). Black trace shows model accuracy differences (r) attributable to the interaction contrast 12–3 channels (Mismatch) – 12–3 channels (Match). Red and blue traces show the contrast 12–3 channels separately for Mismatch and Match conditions, respectively. Shading around each trace represents the standard error of the mean. Horizontal bars at the bottom indicate significant lags for each contrast using the same color scheme as the traces (dark sections indicate p<0.05 FDR corrected across lags and light sections indicate p<0.05 uncorrected). Data have been averaged over the spectrotemporal modulations showing a significant interaction in panel B, indicated also as an inset (top-right).

The online version of this article includes the following figure supplement(s) for figure 6:

**Figure supplement 1.** Timecourse of decoding accuracy (single-lag analysis).

prior knowledge have an interactive influence on these modulation-based representations, peaking approximately 50 ms after speech input: when mismatching text precedes degraded speech, neural representations of spectrotemporal modulations are enhanced with increasing sensory detail whereas the opposite effect is observed when text matches speech. Fourth, this interaction is observed specifically for neural representations of intermediate spectrotemporal modulations that convey speech content. This result stands in marked contrast to what would be expected on the basis of a sharpening scheme in which both signal quality and prior knowledge enhance neural representations, just as they enhance perceptual clarity. An interactive influence of signal quality and prior knowledge is, however, fully consistent with prediction error representations in which neurons signal the difference between expected and heard speech (*Blank and Davis, 2016*).

## Cortical responses encode spectrotemporal modulations in speech

Our finding that MEG responses are driven by spectrotemporal modulations in speech adds to a growing body of evidence implicating spectrotemporal representations as an important component of the neural code by which the cortex represents speech and other sounds. Linguistically relevant features such as formant transitions and speaker fundamental frequency are clearly apparent in these modulations and their removal by filtering has marked consequences for speech intelligibility and speaker discrimination (*Elliott and Theunissen, 2009*; *Flinker et al., 2019*). In animal electrophysiology (*Chi et al., 2005*; *Theunissen and Elie, 2014*), neurons in auditory cortex are shown to be well-tuned to spectrotemporal features in the modulation domain. These findings complement human intracranial (*Pasley et al., 2012*; *Hullett et al., 2016*) and fMRI (*Santoro et al., 2014*; *Santoro et al., 2017*) studies demonstrating that the relationship between auditory stimuli and neural responses is best modelled using spectrotemporal modulations.

Analysis of the acoustic properties of noise-vocoded spoken words show that although low-frequency spectral and temporal modulations dominate the speech signal, only the intermediate modulations (2–4 Hz temporally; 1–2 cycles per octave spectrally) are degraded by noise-vocoding that severely impairs intelligibility (e.g. *Shannon et al., 1995*; *Davis and Johnsrude, 2003*; *Scott et al., 2006*; *Obleser et al., 2008*). These intermediate spectrotemporal modulations encode formant transitions that are an important cue to consonant place and manner of articulation (*Liberman et al., 1967*; *Roberts et al., 2011*). Notably, our decoding analysis shows that it was the intermediate temporal modulations (2–4 Hz) that were preferentially represented in MEG responses. Thus, rather than faithfully encoding the acoustic structure of speech by tracking low-frequency modulations, cortical responses are selectively tuned to acoustic properties (intermediate spectrotemporal modulations) that convey speech information.

Our findings agree with recent work (*Di Liberto et al., 2015*; *Daube et al., 2019*) suggesting that cortical responses to speech (as measured with MEG) are not completely invariant to acoustic

detail. While our analysis that combined different feature spaces also indicate contributions from acoustically invariant phonetic features (consistent with *Di Liberto et al., 2015*), the spectrotemporal modulation feature space best predicted MEG responses when considered individually. Despite the lack of full invariance, spectrotemporal modulations are nonetheless a higher-level representation of the acoustic signal than, for example, the spectrogram. Whereas a spectrogram-based representation would faithfully encode any pattern of spectrotemporal stimulation, a modulation-based representation would be selectively tuned to patterns that change at specific spectral scales or temporal rates. Such selectivity is a hallmark of neurons in higher-levels of the auditory system (*Rauschecker and Scott, 2009*). Our findings are thus consistent with the notion that superior temporal neurons, which are a major source of speech-evoked MEG responses (e.g. *Bonte et al., 2006*; *Sohoglu et al., 2012*; *Brodbeck et al., 2018*), lie at an intermediate stage of the speech processing hierarchy (e.g. *Mesgarani et al., 2014*; *Evans and Davis, 2015*; *Yi et al., 2019*). It should be noted however that in *Daube et al., 2019*, the best acoustic predictor of MEG responses was based on spectral onsets rather than spectrotemporal modulations. It may be that the use of single words in the current study reduced the importance of spectral onsets, which may be more salient in the syllabic transitions during connected speech as used by *Daube et al., 2019*.

Our findings may also reflect methodological differences: the encoding/decoding methods employed here have typically been applied to neural recordings from >60 min of listening to continuous – and clear – speech. By comparison, our stimulus set comprised single degraded words, generating fewer and possibly less varied neural responses, which may limit encoding/decoding accuracy for certain feature spaces. However, we think this is unlikely for two reasons. First, we could replicate several other previous findings from continuous speech studies, for example, greater encoding accuracy for spectrogram versus envelope models (*Di Liberto et al., 2015*) and for the combined spectrogram + phonetic feature model versus spectrogram alone (*Di Liberto et al., 2015*; *Daube et al., 2019*). Second, our effect sizes are comparable to those reported in previous work (e.g. *Di Liberto et al., 2015*) and model comparison results highly reliable, with the spectrotemporal modulation model outperforming the other individual feature spaces in 17 out of 21 participants in the left hemisphere and 18 out of 21 participants in the right hemisphere. Thus, rather than reflecting methodological factors, we suggest that our findings might instead reflect differences in the acoustic, phonetic, and lexical properties of single degraded words and connected speech – including differences in speech rate, co-articulation, segmentation, and predictability. Future work is needed to examine whether and how these differences are reflected in neural representations.

## Prediction error in speech representations

Going beyond other studies, however, our work further shows that neural representations of speech sounds reflect a combination of the acoustic properties of speech, and prior knowledge or expectations. Our experimental approach – combining manipulations of sensory detail and matching or mismatching written text cues – includes specific conditions from previous behavioral (*Sohoglu et al., 2014*) and neuroimaging (*Sohoglu et al., 2012*; *Blank and Davis, 2016*; *Sohoglu and Davis, 2016*; *Blank et al., 2018*) studies that were designed to distinguish different computations for speech perception. In particular, neural representations that show an interaction between sensory detail and prior knowledge provide unambiguous support for prediction error over sharpening coding schemes (*Blank and Davis, 2016*). This interaction has previously been demonstrated for multivoxel patterns as measured by fMRI (*Blank and Davis, 2016*). The current study goes beyond this previous fMRI work in establishing the latency at which prediction errors are computed. As prediction errors reflect the outcome of a comparison between top-down and bottom-up signals, it has been unclear whether such a computation occurs only at late latencies and plausibly results from re-entrant feedback connections that modulate late neural responses (*Lamme and Roelfsema, 2000*; *Garrido et al., 2007*). However, our decoding analysis suggests that the neural interaction between sensory detail and prior knowledge that we observe is already apparent within 100 ms of speech input. This suggests that prediction errors are computed during early feedforward sweeps of processing and are tightly linked to ongoing changes in speech input.

One puzzling finding from the current study is that the interaction between prior knowledge and sensory detail on neural representations is also present when randomly shuffling the spectrotemporal modulation features of different words. While an interaction is still present after controlling for contributions from the shuffled data – albeit only for speech with 3 to 6 channels of sensory detail –

our findings suggest that between-condition differences in model accuracy are also linked to a generic spectrotemporal representation common to all the words in our study. However, it is unclear how differences in encoding accuracy between matching and mismatching conditions could result from a generic representation (since the same words appeared in both conditions). One possibility is that listeners can rapidly detect mismatch and use this to suppress or discard the representation of the written word in mismatch trials. That is, they switch from a high-precision (written text) prior, to a lower precision prior midway through the perception of a mismatching word (see *Cope et al., 2017* for evidence consistent with flexible use of priors supported by frontal and motor speech areas). However, the timecourse of when speech can be decoded in the present study appears inconsistent with this explanation since the neural interaction is present already at early time lags. In this analysis, we focussed on neural responses that track the speech signal with shorter or longer delays. Future research to link neural processing with specific moments in the speech signal (e.g. comparing mismatch at word onset versus offset; *Sohoglu et al., 2014*; *Blank et al., 2018*) would be valuable and might suggest other ways of assessing when and how priors are dynamically updated during perception.

Previous electrophysiological work (*Holdgraf et al., 2016*; *Di Liberto et al., 2018b*) has also shown that matching prior knowledge, or expectation can affect neural responses to speech and representations of acoustic and phonetic features. However, these findings were observed under a fixed degradation level, close to our lowest-clarity, 3-channel condition (see also *Di Liberto et al., 2018a*). Thus, it is unclear whether the enhanced neural representations observed in these earlier studies reflect computations of sharpened signals or prediction errors – both types of representational scheme can accommodate the finding that prior knowledge enhances neural representations for low clarity speech (see *Figure 1D*). By manipulating degradation level alongside prior knowledge, we could test for the presence of interactive effects that distinguish neural representations of prediction errors from sharpened signals (*Blank and Davis, 2016*). This has potential implications for studies in other domains. For example, multivariate analysis of fMRI (*Kok et al., 2012*) and MEG (*Kok et al., 2017*) signals has shown that visual gratings with expected orientations are better decoded than unexpected orientations. In these previous studies, however, the visual properties and hence perceptual clarity of the decoded gratings was held constant. The current study, along with fMRI results from *Blank and Davis, 2016*, suggests that assessing the interaction between expectation and perceptual clarity more clearly distinguishes prediction error from sharpened signal accounts.

Enhanced prediction error representations of low-clarity speech when prior expectations are *more accurate* might seem counterintuitive. In explaining this result, we note that in our study, listeners were able to make strong (i.e. precise) predictions about upcoming speech: Matching and mismatching trials occurred equally frequently and thus upon seeing a written word, there was a. 5 probability of hearing the same word in spoken form. However, in the low-clarity condition, acoustic cues were by definition weak (imprecise). Therefore, in matching trials, listeners would have made predictions for acoustic cues that were absent in the degraded speech input, generating negative prediction errors (shown as dark 'clay' in *Figure 1C*). Critically, however, these negative prediction errors overlap with the cues that are normally present in clear speech (shown as light 'clay' in *Figure 1C*). This close correspondence between negative prediction error and degraded speech cues would result in good encoding/decoding accuracy. In mismatching trials, negative prediction errors would also be generated (shown as dark 'fast' in *Figure 1C*) but would necessarily mismatch with heard speech cues. Thus, while the overall magnitude of prediction error might be highest in this condition (see *Figure 1—figure supplement 1*), there would be limited overlap between predicted and heard speech cues, resulting in worse model accuracy. In summary then, an important factor in explaining our results is the relative precision of predictions and the sensory input.

Other work has taken a different approach to investigating how prior knowledge affects speech processing. These previous studies capitalized on the fact that in natural speech, certain speech sounds are more or less predictable based on the preceding sequence of phonemes (e.g. upon hearing 'capt-", '-ain' is more predictable than '-ive' because 'captain' is a more frequent English word than 'captive'). Using this approach, MEG studies demonstrate correlations between the magnitude of neural responses and the magnitude of phoneme prediction error (i.e. phonemic surprisal, *Brodbeck et al., 2018*; *Donhauser and Baillet, 2020*). However, this previous work did not test for sharpened representations of speech; leaving open the question of whether expectations are

combined with sensory input by computing prediction errors or sharpened signals. Indeed, this question may be hard to address using natural listening paradigms since many measures of predictability are highly correlated (e.g. phonemic surprisal and lexical uncertainty, *Brodbeck et al., 2018*; *Donhauser and Baillet, 2020*). Here by experimentally manipulating listeners' predictions and the quality of sensory input, we could test for the statistical interaction between prior knowledge and sensory detail that most clearly distinguishes prediction errors from sharpened representations. Future work in which this approach is adapted to continuous speech could help to establish the role of predictive computations in supporting natural speech perception and comprehension.

The challenge of experimental control with natural listening paradigms may also explain why acoustic representations of words during connected speech listening are enhanced when semantically similar to previous words (*Broderick et al., 2019*). Under the assumption that semantically similar words are also more predictable, this finding might be taken to conflict with the current findings, that is, weaker neural representations of expected speech in the high-clarity condition. However, this conflict can be resolved by noting that semantic similarity and word predictability are only weakly correlated and furthermore may be neurally dissociable (see *Frank and Willems, 2017*, for relevant fMRI evidence). Further studies that monitor how prior expectations modulate representations concurrently at multiple levels (e.g. acoustic, lexical, semantic) would be valuable to understand how predictive processing is coordinated across the cortical hierarchy.

Our evidence for prediction error representations challenges models, such as TRACE (*McClelland and Elman, 1986*; *McClelland et al., 2014*), in which bottom-up speech input can only ever be sharpened by prior expectations. Instead, our findings support predictive coding accounts in which prediction errors play an important role in perceptual inference. However, it should be noted that in these predictive coding accounts, prediction errors are not the end goal of the system. Rather, they are used to update and sharpen predictions residing in separate neural populations. It is possible that these sharpened predictions are not apparent in our study because they reside in deeper cortical layers (*Bastos et al., 2012*) to which the MEG signal is less sensitive (*Hämäläinen et al., 1993*) or in higher hierarchical levels (e.g. lexical/semantic versus the acoustic level assessed here). It has also been proposed that prediction errors and predictions are communicated via distinct oscillatory channels (e.g. *Arnal et al., 2011*; *Bastos et al., 2012*), raising the possibility that other neural representations could be revealed with time-frequency analysis methods. In future, methodological advances (e.g. in layer-specific imaging; *Kok et al., 2016*) may help to resolve this issue.

We also consider how our results relate to the 'opposing process' account recently proposed by *Press et al., 2020*. According to this account, perception is initially biased by expected information. However, if the stimulus is sufficiently surprising, then later processing is redeployed toward unexpected inputs. Importantly in the context of the current study, this latter process is proposed to operate only when sensory inputs clearly violate expectations, that is, when sensory evidence is strong. While this account is formulated primarily at the perceptual level, a possible implication is that our interaction effect results from the conflation of two distinct processes occurring at different latencies: an early component in which neural responses are upweighted for expected input (leading to greater model accuracy for matching versus mismatching trials) and a later component triggered by high-clarity speech whereby neural responses are upweighted for surprising information (leading to greater model accuracy in mismatching trials). However, the timecourse of when high-clarity speech can be decoded is inconsistent with this interpretation: for high-clarity speech, decoding is only ever lower in matching trials, and never in the opposite direction, as would be expected for neural responses that are initially biased for predicted input (see *Figure 6—figure supplement 1*). It may be that the opposing process model is more applicable to neural representations of predictions rather than the prediction errors that are hypothesized to be apparent here. Future research with methods better suited to measuring neural representations of predictions (e.g. time-frequency analysis, as mentioned above) may be needed to provide a more definitive test of the opposing process model.

# Materials and methods

## Participants

21 (12 female, 9 male) right-handed participants were tested after being informed of the study's procedure, which was approved by the Cambridge Psychology Research Ethics Committee (reference number CPREC 2009.46). All were native speakers of English, aged between 18 and 40 years (mean = 22, SD = 2) and had no history of hearing impairment or neurological disease based on self-report.

## Spoken stimuli

468 monosyllabic words were presented to each participant in spoken or written format drawn randomly from a larger set of monosyllabic spoken words. The spoken words were 16-bit, 44.1 kHz recordings of a male speaker of southern British English and their duration ranged from 372 to 903 ms (mean = 591, SD = 78). The amount of sensory detail in speech was varied using a noise-vocoding procedure (*Shannon et al., 1995*), which superimposes the temporal-envelope from separate frequency regions in the speech signal onto white noise filtered into corresponding frequency regions. This allows parametric variation of spectral detail, with increasing numbers of channels associated with increasing intelligibility. Vocoding was performed using a custom Matlab script (The MathWorks Inc), using 3, 6, or 12 spectral channels spaced between 70 and 5000 Hz according to Greenwood's function (*Greenwood, 1990*). Envelope signals in each channel were extracted using half-wave rectification and smoothing with a second-order low-pass filter with a cut-off frequency of 30 Hz. The overall RMS amplitude was adjusted to be the same across all audio files.

Each spoken word was presented only once in the experiment so that unique words were heard on all trials. The particular words assigned to each condition were randomized across participants. Before the experiment, participants completed brief practice sessions each lasting approximately 5 min that contained all conditions but with a different corpus of words to those used in the experiment. Stimulus delivery was controlled with E-Prime 2.0 software (Psychology Software Tools, Inc).

## Procedure

Participants completed a modified version of the clarity rating task previously used in behavioral and MEG studies combined with a manipulation of prior knowledge (*Davis et al., 2005*; *Hervais-Adelman et al., 2008*; *Sohoglu et al., 2012*). Speech was presented with 3, 6, or 12 channels of sensory detail while prior knowledge of speech content was manipulated by presenting mismatching or matching text before speech onset (see *Figure 1A*). Written text was composed of black lowercase characters presented for 200 ms on a gray background. Mismatching text was obtained by permuting the word list for the spoken words. As a result, each written word in the Mismatching condition was also presented as a spoken word in a previous or subsequent trial and vice versa.

Trials commenced with the presentation of a written word, followed 1050 (±0–50) ms later by the presentation of a spoken word (see *Figure 1A*). Participants were cued to respond by rating the clarity of each spoken word on a scale from 1 ('Not clear') to 4 ('Very clear') 1050 (±0–50) ms after speech onset. The response cue consisted of a visual display of the rating scale and responses were recorded using a four-button box from the participant's right hand. Subsequent trials began 850 (±0–50) ms after participants responded.

Manipulations of sensory detail (3/6/12 channel speech) and prior knowledge of speech content (Mismatching/Matching) were fully crossed, resulting in a 3 × 2 factorial design with 78 trials in each condition. Trials were randomly ordered during each of three presentation blocks of 156 trials.

## Schematic illustrations of sharpened signal and prediction error models

To illustrate the experimental predictions for the sharpened signal and prediction error models, we generated synthetic representational patterns of speech (shown in *Figure 1C*). To facilitate visualization, speech was represented as pixel-based binary patterns (i.e. written text) corresponding to the words 'clay' and 'fast'. To simulate our experimental manipulation of sensory detail, we used a two-dimensional filter to average over eight ('low' sensory detail) or four ('medium') local pixels in the speech input patterns. To simulate the 'high' sensory detail condition, the input patterns were left unfiltered. Patterns for the speech input and predictions were added to uniformly distributed noise

(mean = 0, standard deviation = 0.5) and normalized to sum to one. Sharpened signal patterns were created by multiplying the input and prediction patterns and normalising to sum to one. The prediction error patterns were created by subtracting the predictions from the input.

To approximate the model accuracy measure in our MEG analysis, we correlated a clear (noise-free) representation of the speech input ('clay') with the sharpened signal or prediction error patterns (shown in *Figure 1D* as the squared Pearson's correlation R2). In addition to this correlation metric focusing on representational content, we also report the overall magnitude of prediction error by summing the absolute prediction error over pixels (shown in *Figure 1—figure supplement 1*).

## Data acquisition and pre-processing

Magnetic fields were recorded with a VectorView system (Elekta Neuromag, Helsinki, Finland) containing two orthogonal planar gradiometers at each of 102 positions within a hemispheric array. Data were also acquired by magnetometer and EEG sensors. However, only data from the planar gradiometers were analyzed as these sensors have maximum sensitivity to cortical sources directly under them and therefore are less sensitive to noise artifacts (*Hämäläinen, 1995*).

MEG data were processed using the temporal extension of Signal Source Separation (*Taulu et al., 2005*) in Maxfilter software to suppress noise sources, compensate for motion, and reconstruct any bad sensors. Subsequent processing was done in SPM8 (Wellcome Trust Centre for Neuroimaging, London, UK), FieldTrip (Donders Institute for Brain, Cognition and Behaviour, Radboud University Nijmegen, the Netherlands) and NoiseTools software (http://audition.ens.fr/adc/NoiseTools) implemented in Matlab. The data were subsequently highpass filtered above 1 Hz and downsampled to 80 Hz (with anti-alias lowpass filtering).

## Linear regression

We used ridge regression to model the relationship between speech features and MEG responses (*Pasley et al., 2012*; *Ding and Simon, 2013*; *Di Liberto et al., 2015*; *Holdgraf et al., 2017*; *Brodbeck et al., 2018*). Before model fitting, the MEG data were epoched at matching times as the feature spaces (see below), outlier trials removed and the MEG time-series *z*-scored such that each time-series on every trial had a mean of zero and standard deviation of 1. We then used the mTRF toolbox (*Crosse et al., 2016*; https://sourceforge.net/projects/aespa) to fit two types of linear model.

The first linear model was an encoding model that mapped from the stimulus features to the neural time-series observed at each MEG sensor and at multiple time lags:

$$y = Sw + \varepsilon$$

where $y$ is the neural time-series recorded at each planar gradiometer sensor, $S$ is an $N_{samples}$ by $N_{features}$ matrix defining the stimulus feature space (concatenated over different lags), w is a vector of model weights and $\varepsilon$ is the model error. Model weights for positive and negative lags capture the relationship between the speech feature and the neural response at later and earlier timepoints, respectively. Encoding models were fitted with lags from −100 to 300 ms but for model prediction purposes (see below), we used a narrower range of lags (0 to 250 ms) to avoid possible artefacts at the earliest and latest lags (*Crosse et al., 2016*).

Our encoding models tested four feature spaces (shown in *Figure 2*), derived from the original clear versions of the spoken stimuli (i.e. before noise-vocoding):

1. Broadband envelope, a simple acoustic-based representation of the speech signal, which captured time-varying acoustic energy in a broad 86–4878 Hz frequency range. Thus, this feature space contained information only about the temporal (and not spectral) structure of speech, which alone can cue many speech sounds (e.g. 'ch' versus 'sh' and 'm' versus 'p'; *Rosen, 1992*). This feature space was obtained by summing the envelopes across the spectral channels in a 24-channel noise-vocoder. The center frequencies of these 24 spectral channels were Greenwood spaced: 86, 120, 159, 204, 255, 312, 378, 453, 537, 634, 744, 869, 1011, 1172, 1356, 1565, 1802, 2072, 2380, 2729, 3127, 3579, 4093, and 4678 Hz (see *Greenwood, 1990*). Envelopes were extracted using half-wave rectification and low-pass filtering at 30 Hz.
2. Spectrogram, which was identical to the Envelope feature space but with the 24 individual spectral channels retained. Because this feature space was spectrally resolved, it contained

more information about the identity of speech sounds than the broadband envelope (e.g. fricatives such as 'f' versus 's' and stop consonants such as 'b' versus 'd'; *Liberman et al., 1967*).

3. Spectrotemporal modulations captured regular fluctuations in energy across the frequency and time axes of the spectrogram. This feature space can be considered an abstraction of the spectrogram that more directly indexes perceptually relevant spectrotemporal features, such as spectrally regular harmonics in vowel sounds or fast temporal modulations of stop consonants. Whereas the spectrogram might be appropriate for modelling subcortical processing, it has been proposed that spectrotemporal modulations are a better model of neural representation in auditory cortex (*Chi et al., 2005*; *Theunissen and Elie, 2014*). We used the NSL toolbox in Matlab (http://nsl.isr.umd.edu/downloads.html) to first compute a 128-channel 'auditory' spectrogram (with constant Q and logarithmic center frequencies between 180 and 7040 Hz), before filtering with 2D wavelet filters tuned to spectral modulations of 0.5, 1, 2, 4, and 8 cycles per octave and temporal modulations of 1, 2, 4, 8, and 16 Hz. All other parameters were set to the default (frame length = 8 ms, time constant = 8 ms and no nonlinear compression). Note that temporal modulations can be positive or negative, reflecting the direction of frequency sweeps (i.e. upward versus downward; *Elliott and Theunissen, 2009*). This default representation is a very high-dimensional feature space: frequency × spectral modulation × temporal modulation × temporal modulation direction with 128 × 5×5 × 2 = 6400 dimensions, each represented for every 8 ms time sample in the speech file. We, therefore, averaged over the 128 frequency channels and positive- and negative-going temporal modulation directions. The resulting feature space was a time-varying representation of spectrotemporal modulation content comprised of 25 features (five spectral modulations x five temporal modulations).

4. Phonetic features. The final feature space comprised 13 phonetic features describing the time-varying phonetic properties of speech including vowel backness, voicing, place, and manner of articulation. We first used a forced-alignment algorithm included as part of BAS software (*Kisler et al., 2017*; https://clarin.phonetik.uni-muenchen.de/BASWebServices/interface/Web-MAUSGeneral) to estimate the onset time of each phoneme. We then converted phoneme representations into the following articulatory phonetic features: Voiced, Unvoiced, Bilabial, Labiodental/Dental, Alveolar, Velar, Plosive, Nasal, Fricative, Liquid, Front, Central, and Back (*International Phonetic Association, 1999*). Between the onset time of each phoneme and the onset time of the subsequent phoneme, the corresponding phonetic features were set to one and non-corresponding phonetic features to 0. Phonetic features that occurred infrequently in our stimulus set were excluded or merged (labiodental and dental features). For dipthong vowels and affricate consonants, we averaged the feature vectors for the component speech sounds.

In addition to the four feature spaces above, we also tested more complex combinations of feature spaces that previous work has shown to be good predictors of neural responses. The first of these combined the spectrogram feature space above with the phonetic feature space (*Di Liberto et al., 2015*). The second combined the spectrogram with the half-wave rectified derivative of the spectrogram to capture spectral onsets (*Daube et al., 2019*).

All feature spaces were downsampled to 80 Hz to match the MEG sampling rate and the acoustic feature spaces (i.e. 1, 2, 3 above) were *z*-score transformed such that each time-series on every trial had a mean of zero and standard deviation of 1. For each trial/word, all feature spaces were zero-padded before speech onset to match the duration of the longest negative lag used for model fitting (i.e. 100 ms). Similarly, zero-padding was added after speech offset to match the duration of the longest positive lag (i.e. 300 ms). This enabled information at the beginning and end of each spoken word to inform the model fits for negative and positive lags, respectively.

The second linear model was a decoding model that mapped in the opposite direction from the neural time-series back to speech spectrotemporal modulations (since analysis of encoding models showed this feature space to be most predictive of neural responses). Before model fitting, to remove redundant dimensions and reduce computation time, the data from the 204 planar gradiometers were transformed into principal components and the first 50 components retained. The model was therefore as follows:

$$s = Yw + \varepsilon$$

Here $s$ is a vector expressing the time-varying spectrotemporal modulations (one vector for each of the 25 spectrotemporal modulations), $Y$ is an $N_{samples}$ by $N_{components}$ matrix of MEG data

(concatenated over different lags), $w$ is a vector of model weights and $\varepsilon$ is the model error. Decoding models were fitted with lags from −100 to 300 ms and from 0 to 250 ms for model prediction purposes. As with the encoding analysis, here model weights for positive and negative lags capture the relationship between the speech feature and the neural response at later and earlier timepoints, respectively. Note that this notation may differ to that used in other work, where instead negative lags are used to denote the post-stimulus period for decoding analyses (see *Crosse et al., 2016*). The single-lag analysis proceeded similarly except that model prediction was conducted separately for each lag, generating model accuracies for all lags between −50 and 250 ms.

To control for overfitting, we used ridge regression and varied the lambda parameter (over 17 values as follows: $2^0$, $2^1$, $2^3$, ... $2^{20}$) which varies the degree to which strongly positive or negative weights are penalized (*Crosse et al., 2016*). This lambda parameter was optimized using a leave-one-trial-out cross-validation procedure. For each feature space, sensor, condition, and participant, we fitted the models using data from all but one trial. We then averaged the model weights across trials and used the result to predict the MEG response (for the encoding models) or speech spectro-temporal modulations (for the decoding model) of the left-out trial. We computed model accuracy as the Pearson correlation between the predicted and observed data and repeated this procedure such that model accuracies were obtained for all trials. Before optimizing lambda, model accuracies were averaged across trials, and also across sensors (encoding models) or modulations (decoding model). The optimal lambda value was then selected as the mode of the model accuracy distribution over participants and conditions (*Holdgraf et al., 2016*). When testing the complex models that combined feature spaces, lambda was optimized for each component feature space independently resulting in a search space of 17 × 17 = 289 lambda values (i.e. 'banded' ridge regression; *Nunez-Elizalde et al., 2019*). When conducting the single-lag decoding analysis, lambda was optimized for each lag separately.

When testing for differences in encoding model accuracies, we averaged model accuracies over trials and over the 20 sensors with the highest model accuracies (in each hemisphere separately). For the decoding analysis, model accuracies were averaged over trials. The resulting data were then entered into repeated measures ANOVAs or *t*-tests to examine differences in model accuracy between feature spaces or conditions.

When testing for above-chance encoding model accuracies, null distributions were created for each condition separately by randomly permuting the feature representations across trials (i.e. shuffling the words in our stimulus set) or within trials (i.e. shuffling the spectrotemporal modulation channels and time-bins for each word). This was repeated for 100 permutations using the same ridge regression lambda that was optimized for the non-shuffled data. For each of the 100 permutations, we then subjected the data to the same analysis pipeline as for the non-permuted data (averaging over trials, selecting 20 left-hemisphere and 20 right-hemisphere sensors with the highest model accuracy and averaging over participants). When z-scoring the observed data with respect to the null distributions, the 20 left-hemisphere and 20 right-hemisphere sensors were selected after z-scoring. This ensured the same sensors were used when combining the observed and permuted data to compute the z-scores. z-scores were obtained by taking the observed model accuracy of each condition and participant, subtracting the mean of the relevant null distribution, and dividing by the null distribution standard deviation.

To visualize the weights of the spectrotemporal modulation encoding model, or 'temporal response functions (TRFs)', we used denoising source separation (DSS) to transform the TRFs into a set of linear components (spatial filters) ordered by their consistency over trials (*de Cheveigné and Parra, 2014*). The first three DSS components (i.e. the three most consistent components) were retained and projected back into sensor-space. When applying DSS, the covariance matrices were computed by summing the covariances over features and conditions. Following DSS, we averaged the TRFs across trials and computed the root mean square (RMS) amplitude across all left hemisphere sensors.

## Acknowledgements

This research was supported by the Medical Research Council (Centenary Award to ES and SUAG/044 G101400 to MHD). We are grateful to Maarten van Casteren, Clare Cook, and Lucy MacGregor for their excellent technical support during data collection.

## Additional information

### Funding

| Funder | Grant reference number | Author |
|---|---|---|
| Medical Research Council | SUAG/044 G101400 | Matthew H Davis |
| Medical Research Council | Centenary Award | Ediz Sohoglu |

The funders had no role in study design, data collection and interpretation, or the decision to submit the work for publication.

### Author contributions

Ediz Sohoglu, Conceptualization, Data curation, Formal analysis, Writing - original draft, Writing - review and editing; Matthew H Davis, Conceptualization, Funding acquisition, Writing - review and editing

### Author ORCIDs

Ediz Sohoglu (iD) https://orcid.org/0000-0002-0755-6445
Matthew H Davis (iD) https://orcid.org/0000-0003-2239-0778

### Ethics

Human subjects: Volunteers were tested after informed consent, and consent to publish. The study was approved by the Cambridge Psychology Research Ethics Committee (reference number CPREC 2009.46).

### Decision letter and Author response

Decision letter https://doi.org/10.7554/eLife.58077.sa1
Author response https://doi.org/10.7554/eLife.58077.sa2

## Additional files

### Supplementary files

• Transparent reporting form

### Data availability

Raw data, experimental materials and analysis scripts are available on a public repository (https://osf.io/b2jpt/).

The following dataset was generated:

| Author(s) | Year | Dataset title | Dataset URL | Database and Identifier |
|---|---|---|---|---|
| Sohoglu E | 2020 | Rapid computations of spectrotemporal prediction error support perception of degraded speech | https://osf.io/b2jpt | Open Science Framework, b2jpt |

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
