## [Decision Letter]

**Acceptance summary:**

This paper convincingly demonstrates the importance of prediction errors – the mismatch between what we predict to hear and what we actually hear – for processing speech signals. Theoretical proposals on the role of prediction errors in perception abound, but direct empirical evidence is scarce. Furthermore, the authors show that these prediction errors occur very early on, suggesting that predictions play a fundamental role in sensory processing.

**Decision letter after peer review:**

Thank you for submitting your article "Rapid computations of spectrotemporal prediction error support perception of degraded speech" for consideration by *eLife*. Your article has been reviewed by three peer reviewers, including Peter Kok as the Reviewing Editor and Reviewer #1, and the evaluation has been overseen by Andrew King as the Senior Editor. The following individuals involved in review of your submission have agreed to reveal their identity: Clare Press (Reviewer #2); Edmund C Lalor (Reviewer #3).

The reviewers have discussed the reviews with one another and the Reviewing Editor has drafted this decision to help you prepare a revised submission.

Summary:

This manuscript by Sohoglu and Davis uses sophisticated analysis of previously published MEG data to adjudicate between two models of how prior information might influence the perception of degraded speech. Based on an interesting interaction effect in the data – increasing sensory detail leads to better representation with mismatching prior information, but worse representation for matching prior information – they conclude that the benefits of prior information on degraded speech perception involve neural computations that are consistent with a predictive coding model. They demonstrate that this interaction was present for the first 250 ms of processing. The reviewers found that the manuscript addresses a very interesting question, is well written, and uses a clever design and analyses. The reviewers had questions about the interpretation of the effects that need to be better supported or toned down in a revised manuscript.

1) Consistency of the data with a predictive coding model. It seems that a predictive coding explanation of the reported interaction between sensory detail and prior knowledge depends on two assumptions, and it would strengthen the manuscript to outline these assumptions – assuming the authors agree with them. First, predictive accounts assume two unit types: error units and hypothesis units. It seems that this interaction requires assuming that the error units contribute to the signal to a greater extent than the hypothesis units. Second, it rests upon assuming high precision for the prior distributions. e.g., the authors state "This interaction arises because sensory signals that match prior expectations are explained away more effectively as signal quality increases and hence neural representations are suppressed even as perceptual outcomes improve." This is only true if the prior has high precision. If the prior has low precision then the opposite effect seems more likely. Put another way, if the statistics of one's environment dictate that sight of the word “clay” is followed by low clarity audio of clay, then the 3 channel likely matches the prior to a greater extent than the 12 channel. Is anything known about the statistical structure of the environment with respect to visual prediction of the auditory signal for these words that could support the assumption of high precision on priors?

2) Theoretical positions involving sharpening that would also predict this interaction. We agree that this interaction would not be predicted under a simple sharpening account, but the recent “opposing process” account cited by the authors – which includes sharpening – would make this prediction. This account predicts that sharpening is followed in time by mechanisms that increase the sensory processing of mismatching inputs, but only those generating high KLD. This would only be the case for high quality, not low quality, inputs and therefore would generate this interaction – even if some other particulars would be assumed different. It is worth discussing that the interaction is also consistent with this account, which involves sharpening, even if less consistent with pure sharpening accounts.

3) Regardless of the specific model used to explain the data, one specific aspect of the main interaction effect – as schematized in Figure 1D and as shown in the data in Figure 5A – needs clarification. Why might we expect a bigger response for a low sensory detail stimulus when it is matching rather than non-matching? That is, why are the two leftmost points on the right plot in Figure 1D the way they are? It seems intuitive that the error should be smaller for an imprecise "clay" than for an imprecise "fast", but the data suggest the opposite is true. We wonder if there would there be any complementary value to trying to predict the MEG based on an acoustic representation of the mismatching text? In other words, if I have a strong prediction of "fast" and I hear an imprecise "clay" would I get a smaller response (representing the error for "fast") than if I had heard a precise "clay". We leave it up to the authors to decide whether this analysis is feasible and valuable; in any case, this finding (reduced error for matching vs. mismatching low quality speech), and its interpretation, should be explained more fully.

4) The logic of the study seems to rest upon effects reflecting sensory processing. It seems plausible that higher level regions would also exhibit high model accuracy, especially given that 250 ms will have allowed sensory information to be communicated throughout most of the cortex. It would therefore strengthen the manuscript to increase the evidence that the effect is sensory. This could be achieved through additional analysis e.g., restricting the analysis to a shorter temporal window, if the data allows, or through theoretical reasoning to support the sensory nature of effects, e.g., related to the linear relationship between spectrotemporal modulations and the MEG signal. In any case, given that a lot of (re-entrant) processing occurs in the first 250 ms, the current claims about early processing, especially references to the feedforward sweep and suggestions that re-entrant processing only occurs > 250 ms, should be strongly toned down.

5) The reviewers had a concern about the use of different speech representations with the TRF. First, they thought the analysis of Figure 3 lacked an assessment of a model that combined spectrotemporal modulations and phonetic features. These models both do pretty well on their own, but should also be complementary.

6) A complementary concern was that there may not be enough data – or enough variability in the specific stimuli – to conduct a fair test of the various representations. Some representations have more dimensions than others. And it seems the TRFs were fit using data acquired to 468 words (many of which were presented with very low levels of intelligibility). This might not be enough data to get quality fits for some of the representations and, thus, the results in Figure 3 may not be quite fair. We appreciate what the authors are trying to do in terms of finding a representation that gives them the best TRF predictions and, thus, the best SNR for testing their hypothesis. However, this issue should be discussed somewhere in the manuscript – to make it clear that the results of Figure 3 are based on a fairly limited amount of data and might not necessarily generalize to a larger data set.

7) Broderick et al., 2019 report seemingly opposite effects of prior knowledge to those presented here. In their case prior knowledge results from semantic similarity of preceding spoken words, and they observed enhanced rather than suppressed encoding of matching spoken words. This discrepancy should be discussed.

8) If the speech system preferentially represents inputs that mismatch predictions, how do the authors explain that valid prior knowledge enhances speech intelligibility? This should be discussed.

[Editors' note: further revisions were suggested prior to acceptance, as described below.]

Thank you for resubmitting your work entitled "Rapid computations of spectrotemporal prediction error support perception of degraded speech" for further consideration by *eLife*. Your revised article has been evaluated by Andrew King (Senior Editor) and a Reviewing Editor.

The manuscript has been greatly improved but there are some remaining issues that need to be addressed before acceptance, as outlined below:

1) The new analyses performed in response to point 3 revealed that the main result of the paper, an interaction between match/mismatch and sensory detail, is also present in a null distribution of their data. The authors offer a potential explanation, but admit that they do not fully understand this result. This seems concerning; if the main result of their study is also present in a control analysis with shuffled trial labels, it is unclear how to interpret these effects. Given the impact of this finding on the interpretability of the results, it does not seem appropriate to omit these results from the manuscript. We recommend including these findings in the paper, along with the (admittedly speculative, but plausible) explanation offered by the authors. Note that this refers specifically to the results of the null distribution analysis of the main results, the reviewers agree with the authors that it's fine to omit the more speculative “written mismatch” analysis (i.e., using the acoustic representation of a word they didn't actually hear) suggested by the reviewers in the previous round.

2) The authors point out how they think they obtained their results because MEG is more sensitive to error unit than hypothesis unit processing. If this was a general rule, two visual MEG studies by Kok and colleagues (2017 PNAS; 2020 J Neurosci) would be unlikely to have found these sharpening patterns. Could the authors say anything about this discrepancy? Is it perhaps more plausible that these auditory data, in combination with the visual findings, might suggest a difference between visual and auditory systems in the extent to which MEG reflects hypothesis v error processing?

3) Relatedly, the authors have interestingly added a time resolved analysis. Given the interaction is seen very early, they consider this finding less consistent with the opposing process account (Press et al., 2020). However, the opposing process account describes perception, and in terms of neural processing is proposed to be reflected in hypothesis unit processing, not error unit processing. If the authors believe they are seeing the processing in error units only in their study, the data do not in fact speak to the opposing process account. If our understanding of the authors' claims is correct, we would suggest amending the paragraph that mentions this account – mentioning that it's a theory of perception and more concerned with hypothesis unit processing which is hypothesised not to be reflected in the present data.

4) The authors have increased clarity concerning how the predictive coding account could explain the present dataset – by clarifying the assumed precision of the priors. However, the reviewers felt the fact that there is greater representation of matching 3 channel than non-matching 3 channel under this account could still be explained more clearly. It was not fully clear how the clarifications provided speak to this particular point. E.g, if assuming a precise gaussian prediction for clay, an imprecise gaussian with the same mean (clay) will overlap with that gaussian more than an imprecise gaussian with a different mean (fast). Higher Kullback Leibler divergence for mismatch still i.e., by prescribing the shape of the prior, they have accounted for why 3 channel can generate more error than 12 channel, but not relative differences between match and non-match in the 3 channel. We think it would help readers if the logic could be further clarified here as our conceptualisation of error processing may deviate from the authors'.

5) With respect to the discussion of the lack of correspondence to the Broderick findings – this may be due to the fact that the Broderick paper was based on semantic similarity – which is not the same thing as predictability. Indeed, Frank and Willems (2017) found that semantic similarity and semantic surprisal were not strongly correlated with each other. It may even be that the Broderick results based on semantic similarity map better onto the earlier activity from hypothesis units and do not tap into error unit activity as much (which might be accessible by exploring the top-down role of semantic surprisal).

6) We don't insist on making changes necessarily – but the transition from the "Stimulus feature space selection" subsection to the "Acoustic analysis…" subsection (p12 – 13) still seems a little weak. You have just spent a lot of time comparing models and then quite quickly converge on the spectrotemporal one. You attempt to justify it might best match with the vocoded speech – but might not that actually work against you? The idea is that you are hoping to be sensitive to error, no? So if you have a strong prior and then get a degraded speech input, don't you want to be sensitive to the details of that prior – and, hence, the error. Again, we don't insist on any major change here. It may suffice that you just wanted to keep the story fairly straightforward and so wanted to pick the single speech representation that performed best.

---

## [Author Response]

Revisions for this paper:1) Consistency of the data with a predictive coding model. It seems that a predictive coding explanation of the reported interaction between sensory detail and prior knowledge depends on two assumptions, and it would strengthen the manuscript to outline these assumptions – assuming the authors agree with them. First, predictive accounts assume two unit types: error units and hypothesis units. It seems that this interaction requires assuming that the error units contribute to the signal to a greater extent than the hypothesis units. Second, it rests upon assuming high precision for the prior distributions. e.g., the authors state "This interaction arises because sensory signals that match prior expectations are explained away more effectively as signal quality increases and hence neural representations are suppressed even as perceptual outcomes improve." This is only true if the prior has high precision. If the prior has low precision then the opposite effect seems more likely. Put another way, if the statistics of one's environment dictate that sight of the word “clay” is followed by low clarity audio of clay, then the 3 channel likely matches the prior to a greater extent than the 12 channel. Is anything known about the statistical structure of the environment with respect to visual prediction of the auditory signal for these words that could support the assumption of high precision on priors?

We agree that the consistency of our data with a predictive coding account depends on these two assumptions.

In the previous version of the manuscript, we briefly mentioned the first assumption. In the revision, we now outline the assumption in greater detail:

“Our evidence for prediction error representations challenges models, such as TRACE (McClelland and Elman, 1986; McClelland et al., 2014), in which bottom-up speech input can only ever be sharpened by prior expectations. […] In future, methodological advances (e.g. in layer-specific imaging; Kok et al., 2016) may help to resolve this issue.”

We also now outline the second assumption in the Discussion section. We believe that outlining this second assumption also helps to address comment #3 below i.e. clarifying why the interaction between prior knowledge and sensory detail comes about.

“Enhanced prediction error representations of low-clarity speech when prior expectations are more accurate might seem counterintuitive. […] In summary then, an important factor in explaining our results is the relative precision of predictions and the sensory input.”

2) Theoretical positions involving sharpening that would also predict this interaction. We agree that this interaction would not be predicted under a simple sharpening account, but the recent “opposing process” account cited by the authors – which includes sharpening – would make this prediction. This account predicts that sharpening is followed in time by mechanisms that increase the sensory processing of mismatching inputs, but only those generating high KLD. This would only be the case for high quality, not low quality, inputs and therefore would generate this interaction – even if some other particulars would be assumed different. It is worth discussing that the interaction is also consistent with this account, which involves sharpening, even if less consistent with pure sharpening accounts.

We now discuss the opponent process model in relation to our results in the Discussion section:

“We also consider a final possibility: that sharpened signals and prediction errors are apparent at different times relative to stimulus input (Press et al., 2020). […] Future research to link neural processing with specific moments in the speech signal (e.g. comparing mismatch at word onset versus offset; Sohoglu et al., 2014; Blank et al., 2018) would be valuable and might suggest other ways of assessing the opponent process account.”

3) Regardless of the specific model used to explain the data, one specific aspect of the main interaction effect – as schematized in Figure 1D and as shown in the data in Figure 5A – needs clarification. Why might we expect a bigger response for a low sensory detail stimulus when it is matching rather than non-matching? That is, why are the two leftmost points on the right plot in Figure 1D the way they are? It seems intuitive that the error should be smaller for an imprecise "clay" than for an imprecise "fast", but the data suggest the opposite is true.

We hope that our response above to comment #1, and the associated changes to the manuscript, clarify why prediction error *representations* should be enhanced for low-clarity speech when (strong) expectations are matching versus mismatching. We emphasize *representations* here as the interaction between sensory detail and prior knowledge that distinguishes prediction error and sharpening models is apparent for multivariate (and not univariate) measures (Blank and Davis, 2016). Thus, according to a prediction error account, we should expect a more informative response (but not necessarily a “bigger” response or an increase in the overall magnitude of perdiction error) for a low sensory detail stimulus when it is matching rather than mismatching.

We wonder if there would there be any complementary value to trying to predict the MEG based on an acoustic representation of the mismatching text? In other words, if I have a strong prediction of "fast" and I hear an imprecise "clay" would I get a smaller response (representing the error for "fast") than if I had heard a precise "clay". We leave it up to the authors to decide whether this analysis is feasible and valuable;

Following Figure 1C, we might in fact expect the opposite pattern i.e. the acoustic representation of mismatching text could be most apparent for low-clarity speech, under both prediction error and sharpened representational schemes. Note that even if this is true, this analysis is less valuable for distinguishing between accounts, since in this case both representational schemes make similar experimental predictions.

We are also not confident about the feasibility of this analysis. Acoustic information is dynamic and unfolds over time. It is therefore unclear how best to timelock neural responses to dynamic acoustic predictions from mismatching text that are never physically present.

Nonetheless, we have performed the analysis and show the results below. Here “(written)Mismatch” is a new condition in which neural encoding accuracy is assessed for the acoustic (spectrotemporal modulation) representation of the mismatching written word. Here we make the simplifying assumption that the acoustic representation of the written word is apparent at the same time as heard speech, and follows the exact same timecourse as if the spoken word were present. Both of these assumptions might be false – if for instance, it might be that only predictions for the initial portion of the word are expressed, or that predictions are pre-played at a different rate from the spoken word itself.

As can be seen in Author response image 1, encoding accuracy in the (written)Mismatch condition increases with increasing speech clarity i.e. counter to what may be suggested in Figure 1C on the basis of either prediction error or sharpening accounts.

**Author response image 1. sa2fig1:** Encoding accuracy in left hemisphere sensors. Data have been analyzed in an identical fashion to Figure 5A except that we show a new (written)Mismatch condition in which the spectrotemporal modulation representation of the written word is used to predict neural responses. There was a significant main effect of sensory detail for this (written)Mismatch condition (F(2,40) = 4.466 p = .023). However, as explained below, permutation testing in which we shuffled the order of the stimuli (over 100 permutations) revealed that each (written)Mismatch datapoint did not significantly differ from the permutation null distribution. ns p >=.05. This is in contrast to the match/mismatch conditions reported in the manuscript which all showed above-chance encoding accuracy for the spoken word that was heard irrespective of sensory clarity.

Puzzled by this, we then, compared encoding accuracy to a null distribution. This null distribution was created by randomly permuting the feature representations across trials (i.e. shuffling the words in our stimulus set), for each condition separately, and repeating this for 100 permutations (but always using the same ridge regression lambda that was previously found to be optimal for the non-shuffled data). For each of the 100 permutations, we then subjected the data to the same analysis pipeline as for the non-permuted data (averaging over trials, selecting 20 left-hemisphere sensors with the highest model accuracy and averaging over participants). This resulted in the null distributions shown in Author response image 2:

**Author response image 2. sa2fig2:** Histograms showing the number of permutations (random shuffles of feature space across trials) as a function of (binned) model accuracy when predicting the MEG response from the spectrotemporal modulation representation of mismatching text. Vertical blue lines indicate the observed (non-permuted) model accuracies.

Importantly, encoding accuracy for (written)Mismatch is never above chance, as defined by this null distribution. It is therefore difficult to interpret between-condition differences in encoding accuracy for this (written)Mismatch condition. However, we also noticed that the null distributions appear to shift depending on experimental condition – i.e. the null distribution of model accuracy increased for written(Mismatch) when higher-clarity speech is presented.

To follow-up on this, we also constructed null distributions for the original six conditions i.e. we predicted the MEG response from a spectrotemporal modulation representation of mismatching and matching speech, presented with 3/6/12 channels of sensory detail. Here we denote prior knowledge conditions as (speech)Mismatch and speech(Match) to distinguish these conditions from the new (written)Mismatch condition presented above. As before, we constructed a null distribution based on 100 permutations in which we randomly assigned spectrotemporal modulation representations for different words to different MEG trials within each condition.

Reassuringly, model accuracies in these conditions are well above chance as defined by the null distribution (*p* <.01 for all six conditions). Hence, in both match and mismatch conditions, neural responses encode the specific acoustic form of the spoken word that is heard. Surprisingly, however, a match/mismatch by sensory detail interaction is also apparent in the null distributions, as shown in Author response image 3:

**Author response image 3. sa2fig3:** Histograms showing the number of permutations (random shuffles of feature space across trials) as a function of (binned) model accuracy when predicting the MEG response from the spectrotemporal modulation representation of mismatching/matching speech. Vertical blue lines indicate the observed (non-permuted) model accuracies.

We do not fully understand this result. Our preferred explanation is that a spectrotemporal representation of a generic or randomly chosen word can also predict MEG responses with some degree of accuracy. This is possible because the words used in the experiment have homogeneous acoustic properties – e.g. they are all monosyllabic words spoken by the same individual. All words therefore share, to some extent, a common spectrotemporal profile. By this interpretation, then, the result in Author response image 3 again confirms that encoding of this shared spectrotemporal profile in MEG responses depends on the combination of (mis)match with prior knowledge and sensory detail.

To help confirm that other simple properties of the neural response cannot explain the observed model accuracies (e.g. differences in the mean univariate response over conditions), we also confirmed that we obtain near-zero encoding accuracy for a fully-permuted model. That is, randomly permuting spectrotemporal modulation channels and time-bins gives us the results shown in Author response image 4:

**Author response image 4. sa2fig4:** Histograms showing permutation null distributions when predicting the MEG response from the spectrotemporal modulation representation of mismatching/matching speech but now for a fully-permuted model (randomly permuting spectrotemporal modulation channels and time-bins). Note the scale for these graphs is much smaller than the previous figures (-.002 to.002 versus.095 to.14), reflecting the substantial drop in model accuracy to near-zero for the fully permuted model.

We therefore feel confident that the interaction between match/mismatch and sensory detail – both in the results shown in Figure 5A of the manuscript, and as shown in Author response image 3 of this letter – supports a prediction error scheme. Encoding of an acoustic representation of a heard word – either for the specific word spoken, or from generic acoustic elements shared with other monosyllabic words from the same speaker – shows an interaction between prior knowledge and sensory detail that is a unique signature of a prediction error scheme. We still do not fully understand why a generic representation would lead to differences in encoding accuracy between match and mismatch conditions. One possibility is that listeners can rapidly detect mismatch and use this to suppress or discard the representation of the written word in mismatch trials. That is, they switch from a high-precision (written text) prior, to a low precision or neutral prior midway through the perception of a mismatching word. However, the time course of the interaction shown in decoding analyses (see point 4 below) would argue against this – there’s no evidence that the neural interaction emerges only at later time points.

In addition, we still do not fully understand why encoding accuracy for (written)Mismatch as shown in Author response image 1 was never above chance levels. At face value, both a prediction error and a sharpening scheme would lead us to expect this outcome – at least for speech that is sufficiently degraded. Indeed it has previously been shown that expected sensory stimuli can be decoded from stimulus omissions (e.g. Kok, Failing and de Lange, 2014, J Cogn Neurosci), or from responses to uninformative stimuli (e.g. Wolff et al., 2017, Nat Neurosci). Further studies in which we use other methods (e.g. time-frequency analysis) to detect less timelocked neural representations of predicted speech stimuli would be valuable to address this issue.

We therefore feel that the added value of this written mismatch analysis is limited and have opted not to add these analyses to the manuscript.

We present it here in the hope that you might help us to understand how to interpret these analyses and potentially incorporate these into our manuscript. However, in the absence of a more confident interpretation, or an explanation of how these findings should change our conclusions we would prefer to exclude these from the manuscript.

In any case, this finding (reduced error for matching vs. mismatching low quality speech), and its interpretation, should be explained more fully.

As we mention above, we hope that our response to comment #1, and the associated changes to the manuscript clarify this issue.

4) The logic of the study seems to rest upon effects reflecting sensory processing. It seems plausible that higher level regions would also exhibit high model accuracy, especially given that 250 ms will have allowed sensory information to be communicated throughout most of the cortex. It would therefore strengthen the manuscript to increase the evidence that the effect is sensory. This could be achieved through additional analysis e.g., restricting the analysis to a shorter temporal window, if the data allows, or through theoretical reasoning to support the sensory nature of effects, e.g., related to the linear relationship between spectrotemporal modulations and the MEG signal. In any case, given that a lot of (re-entrant) processing occurs in the first 250 ms, the current claims about early processing, especially references to the feedforward sweep and suggestions that re-entrant processing only occurs > 250 ms, should be strongly toned down.

We agree that a key interpretation of our study is that prediction errors are computed at an early acoustic level involving representations of spectrotemporal modulations. We have therefore followed your suggestion to implement a time-resolved analysis, which we now report in the Results section and in Figure 6 as a new panel D. This analysis suggests that the critical interaction effect between sensory detail and prior knowledge (on spectrotemporal decoding) peaks approximately 50 ms after speech input. This finding is consistent with our interpretation of an early sensory locus.

In addition, we no longer suggest that 250 ms is a latency indicative of feedforward or feedback processing.

We thank you for this helpful suggestion to strengthen our manuscript.

5) The reviewers had a concern about the use of different speech representations with the TRF. First, they thought the analysis of Figure 3 lacked an assessment of a model that combined spectrotemporal modulations and phonetic features. These models both do pretty well on their own, but should also be complementary.

We now include a more comprehensive model comparison that includes the feature space combination of spectrotemporal modulations and phonetic features. This appears as a new Figure 3—figure supplement 1 and new sections of text in the Results section.

The outcome of this new analysis indicates that spectrotemporal modulations+phonetic features outperforms spectrotemporal modulations alone. This suggests that both phonetic features and spectrotemporal modulations are represented in neural responses. We now have modified the Discussion section to reflect these new findings:

“Our findings agree with recent work (Di Liberto et al., 2015; Daube et al., 2019) suggesting that cortical responses to speech (as measured with MEG) are not completely invariant to acoustic detail. While our analysis that combined different feature spaces also indicate contributions from acoustically invariant phonetic features (consistent with Di Liberto et al., 2015), the spectrotemporal modulation feature space best predicted MEG responses when considered individually.”

Importantly, however, this does not negate our previous finding that spectrotemporal modulations is the feature space that individually best explains MEG variance. Spectrotemporal modulations also provides the clearest link to our noise-vocoding manipulation, which degrades speech by removing narrowband spectral modulations. In the revised manuscript, we have made this link more explicit:

“Our analysis above suggests that a feature space comprised of spectrotemporal modulations is most accurately represented in neural responses. […] To investigate the acoustic impact of noise-vocoding on our stimuli, we next characterized the spectrotemporal modulations that convey speech content in our stimulus set and how those modulations are affected by noise-vocoding with a wide range of spectral channels, from 1 to 24 channels.”

6) A complementary concern was that there may not be enough data – or enough variability in the specific stimuli – to conduct a fair test of the various representations. Some representations have more dimensions than others. And it seems the TRFs were fit using data acquired to 468 words (many of which were presented with very low levels of intelligibility). This might not be enough data to get quality fits for some of the representations and, thus, the results in Figure 3 may not be quite fair. We appreciate what the authors are trying to do in terms of finding a representation that gives them the best TRF predictions and, thus, the best SNR for testing their hypothesis. However, this issue should be discussed somewhere in the manuscript – to make it clear that the results of Figure 3 are based on a fairly limited amount of data and might not necessarily generalize to a larger data set.

We acknowledge that because our stimulus set consisted of single words, rather than connected speech, there is a concern *apriori* that there may be not be enough data to distinguish different representations. However, our results indicate that this is not the case because 1) we can replicate several key findings from studies using similar methods and 2) our effect sizes are comparable to those reported in previous work and 3) our model comparison results are highly reliable (shown in Figure 3A).

Nonetheless, we agree that this issue should be acknowledged, and we now do so in the Discussion section:

“Our findings may also reflect methodological differences: the encoding/decoding methods employed here have typically been applied to neural recordings from > 60 minutes of listening to continuous – and undegraded – speech. […] Future work is needed to examine whether and how these differences are reflected in neural representations.”

7) Broderick et al., 2019 report seemingly opposite effects of prior knowledge to those presented here. In their case prior knowledge results from semantic similarity of preceding spoken words, and they observed enhanced rather than suppressed encoding of matching spoken words. This discrepancy should be discussed.

We now discuss this study in the Discussion section:

“The challenge of experimental control with natural listening paradigms may also explain why accurate semantic predictions appear to enhance acoustic representations during connected speech listening (Broderick et al., 2019) i.e. opposite to the effect we observe for the high-clarity condition shown in Figure 5A. […] In our paradigm the assignment of spoken words to conditions was counterbalanced across participants, allowing us to control for all acoustic differences between matching and mismatching expectation conditions.”

8) If the speech system preferentially represents inputs that mismatch predictions, how do the authors explain that valid prior knowledge enhances speech intelligibility? This should be discussed.

We now clarify this in the Introduction section:

“This pattern – opposite effects of signal quality on neural representations depending on whether prior knowledge is informative or uninformative – highlights an important implication of neural activity that represents prediction errors. Rather than directly signaling perceptual outcomes, these neural signals in sensory cortex serve the intermediary function of updating higher-level (phonological, lexical or semantic) representations so as to generate a perceptual interpretation (the posterior, in Bayesian terms) from a prediction (prior). It is these updated perceptual interpretations, and not prediction errors, that should correlate most closely with perceived clarity (Sohoglu and Davis, 2016).”

[Editors' note: further revisions were suggested prior to acceptance, as described below.]

The manuscript has been greatly improved but there are some remaining issues that need to be addressed before acceptance, as outlined below:1) The new analyses performed in response to point 3 revealed that the main result of the paper, an interaction between match/mismatch and sensory detail, is also present in a null distribution of their data. The authors offer a potential explanation, but admit that they do not fully understand this result. This seems concerning; if the main result of their study is also present in a control analysis with shuffled trial labels, it is unclear how to interpret these effects. Given the impact of this finding on the interpretability of the results, it does not seem appropriate to omit these results from the manuscript. We recommend including these findings in the paper, along with the (admittedly speculative, but plausible) explanation offered by the authors. Note that this refers specifically to the results of the null distribution analysis of the main results, the reviewers agree with the authors that it's fine to omit the more speculative “written mismatch” analysis (i.e., using the acoustic representation of a word they didn't actually hear) suggested by the reviewers in the previous round.

Thank you for this helpful suggestion. We agree that this analysis is important for interpretation and now include it in our revised manuscript as new next in the Results and Discussion sections as well as a new Figure 5—figure supplement 1. We also provide additional analysis in which the observed data are *z*-scored with respect to the feature-shuffled and word-shuffled distributions (Figure 5—figure supplement 1B and D, respectively). Interestingly, this analysis suggests that an interaction between prior knowledge and sensory detail on model accuracy is still present in the data even after controlling for contributions from the word-shuffled distributions. The precise form of the interaction differs to that seen previously: increasing sensory detail has opposite effects on model accuracy when prior knowledge is matching vs mismatching but only for speech ranging from 3 to 6 channels. Nonetheless, this suggests an interaction that favours prediction errors is word-specific to some extent, as originally interpreted. On the other hand, it is also true that a large contribution to the original interaction effect appears to be linked to a generic representation and we now clearly acknowledge this in the revised manuscript (Results and Discussion sections).

2) The authors point out how they think they obtained their results because MEG is more sensitive to error unit than hypothesis unit processing. If this was a general rule, two visual MEG studies by Kok and colleagues (2017 PNAS; 2020 J Neurosci) would be unlikely to have found these sharpening patterns. Could the authors say anything about this discrepancy? Is it perhaps more plausible that these auditory data, in combination with the visual findings, might suggest a difference between visual and auditory systems in the extent to which MEG reflects hypothesis v error processing?

We cannot rule out the possibility that these differences in findings reflect differences in how visual and auditory systems are expressed in MEG signals. However, given that our interaction between prior knowledge and sensory detail is also present in speech fMRI data (Blank and Davis, 2016), we do not think the discrepancy is linked to MEG specifically. This could suggest a more general difference between vision and audition (irrespective of imaging modality). On the other hand, there is other work that has shown decreased decoding accuracy for expected visual stimuli (inconsistent with a sharpening model; Kumar 2017 *JoCN*).

Our preferred explanation is therefore that if visual paradigms were adapted to include a range of perceptual clarity levels (so as to permit a test of the interaction with expectation), sharpening versus prediction error models could be more definitively tested. We now provide further discussion along these lines:

“This has potential implications for studies in other domains. For example, multivariate analysis of fMRI (Kok et al., 2012) and MEG (Kok et al., 2017) signals has shown that visual gratings with expected orientations are better decoded than unexpected orientations. In these previous studies, however, the visual properties and hence perceptual clarity of the decoded gratings was held constant. The current study, along with fMRI results from Blank and Davis (2016), suggests that assessing the interaction between expectation and perceptual clarity more clearly distinguishes prediction error from sharpened signal accounts.”

3) Relatedly, the authors have interestingly added a time resolved analysis. Given the interaction is seen very early, they consider this finding less consistent with the opposing process account (Press et al., 2020). However, the opposing process account describes perception, and in terms of neural processing is proposed to be reflected in hypothesis unit processing, not error unit processing. If the authors believe they are seeing the processing in error units only in their study, the data do not in fact speak to the opposing process account. If our understanding of the authors' claims is correct, we would suggest amending the paragraph that mentions this account – mentioning that it's a theory of perception and more concerned with hypothesis unit processing which is hypothesised not to be reflected in the present data.

Thank you for pointing out this inaccuracy in our description of the opposing process account. We now clarify that the model is primarily formulated at the perceptual level. We still think that our decoding results can be argued to be inconsistent with the model. But we now acknowledge that the model might be more applicable to hypothesis unit processing, rather than the prediction errors we hypothesise are reflected in our data:

“We also consider how our results relate to the ‘opposing process’ account recently proposed by Press et al. (2020). […] Future research with methods better suited to measuring neural representations of predictions (e.g. time-frequency analysis, as mentioned above) may be needed to provide a more definitive test of the opposing process model.”

4) The authors have increased clarity concerning how the predictive coding account could explain the present dataset – by clarifying the assumed precision of the priors. However, the reviewers felt the fact that there is greater representation of matching 3 channel than non-matching 3 channel under this account could still be explained more clearly. It was not fully clear how the clarifications provided speak to this particular point. E.g, if assuming a precise gaussian prediction for clay, an imprecise gaussian with the same mean (clay) will overlap with that gaussian more than an imprecise gaussian with a different mean (fast). Higher Kullback Leibler divergence for mismatch still i.e., by prescribing the shape of the prior, they have accounted for why 3 channel can generate more error than 12 channel, but not relative differences between match and non-match in the 3 channel. We think it would help readers if the logic could be further clarified here as our conceptualisation of error processing may deviate from the authors'.

We believe this point can be clarified by highlighting the distinction between the overall magnitude of neural response and representational content.

We agree that, under a prediction error scheme, there should be higher Kullback Liebler Divergence (KLD) for Mismatch than Match conditions for low-clarity speech. To confirm this, we computed the KLD between the speech input and prediction patterns in Figure 1C. The results are shown in Author response image 5:

**Author response image 5. sa2fig5:** Kullback Liebler Divergence (KLD) between speech input and prediction patterns (shown in Figure 1C).

However, because the KLD measure integrates over all representational units, this measure is effectively capturing the overall magnitude of prediction error. A similar result would be obtained if instead of the KLD, we took the sum of absolute prediction error (cf. Gagnepain et al., 2012 *Current Biology*).

In contrast to analysing the KLD (or summed absolute prediction error), we can instead correlate a clear template pattern of “clay” with the prediction error patterns. This allows us to approximate our main model accuracy measure used in the manuscript i.e. measuring how well the original (clear) spoken word is represented in MEG responses to vocoded versions of the speech signal. Expressing the correlation as the coefficient of determination (R2), gives us the results in Figure 1D.

As can be seen in Figure 1D, when using a measure that captures representational content of heard speech, we can successfully reproduce the qualitative pattern of results in the low clarity condition i.e. better prediction error representation of heard speech when prior expectations match than mismatch.

To clarify this point, we have made the following changes:

Revised Figure 1D so that we now show the R2 measure described above

(instead of the “cartoon” depiction in previous versions)

We include a new figure supplement to Figure 1 to illustrate how the overall magnitude of prediction error is expected to differ between conditions. For this we show the summed absolute prediction error as a simpler alternative to the KLDIn Figure 1C, we now include depictions for three levels of sensory detail for better correspondence with our experimental design (previously we only showed two levels)The changes above are also accompanied by next text in the Materials and methods section to explain how the illustrations and associated metrics (R2, magnitude of prediction error) in Figure 1 and supplemental figure 1 were obtainedWe have modified the caption to Figure 1 to highlight the different outcomes for the R2 and magnitude of prediction error metrics

We have also modified the relevant part of the Discussion:

“Therefore, in matching trials, listeners would have made predictions for acoustic cues that were absent in the degraded speech input, generating negative prediction errors (shown as dark “clay” in Figure 1C). […] Thus, while the overall magnitude of prediction error might be highest in this condition (see Figure 1 —figure supplement 1), there would be limited overlap between predicted and heard speech cues, resulting in worse model accuracy.”

5) With respect to the discussion of the lack of correspondence to the Broderick findings – this may be due to the fact that the Broderick paper was based on semantic similarity – which is not the same thing as predictability. Indeed, Frank and Willems (2017) found that semantic similarity and semantic surprisal were not strongly correlated with each other.

We agree that the lack of correspondence here is likely because semantic similarity is dissociable from word predictability. We have amended this section to clarify this point:

“The challenge of experimental control with natural listening paradigms may also explain why acoustic representations of words during connected speech listening are enhanced when semantically similar to previous words (Broderick et al., 2019). Under the assumption that semantically similar words are also more predictable, this finding might be taken to conflict with the current findings i.e. weaker neural representations of expected speech in the high-clarity condition. However, this conflict can be resolved by noting that semantic similarity and word predictability are only weakly correlated and furthermore may be neurally dissociable (see Frank and Willems, 2017, for relevant fMRI evidence).”

It may even be that the Broderick results based on semantic similarity map better onto the earlier activity from hypothesis units and do not tap into error unit activity as much (which might be accessible by exploring the top-down role of semantic surprisal).

We now acknowledge that an important question for future research is to investigate how prediction is expressed at multiple hierarchical levels, including semantics:

“Further studies that monitor how prior expectations modulate representations concurrently at multiple levels (e.g. acoustic, lexical, semantic) would be valuable to understand how predictive processing is coordinated across the cortical hierarchy.”

While we do not mention hypothesis units in this section, we have modified a later part of the discussion to acknowledge that hypothesis unit processing might be more evident in higher (e.g. semantic) levels than the acoustic level apparent in the current study:

“It is possible that these sharpened predictions are not apparent in our study because they reside in deeper cortical layers (Bastos et al., 2012) to which the MEG signal is less sensitive (Hämäläinen et al., 1993) or in higher hierarchical levels (e.g. lexical/semantic versus the acoustic level assessed here).”

6) We don't insist on making changes necessarily – but the transition from the "Stimulus feature space selection" subsection to the "Acoustic analysis…" subsection (p12 – 13) still seems a little weak. You have just spent a lot of time comparing models and then quite quickly converge on the spectrotemporal one. You attempt to justify it might best match with the vocoded speech – but might not that actually work against you? The idea is that you are hoping to be sensitive to error, no? So if you have a strong prior and then get a degraded speech input, don't you want to be sensitive to the details of that prior – and, hence, the error. Again, we don't insist on any major change here. It may suffice that you just wanted to keep the story fairly straightforward and so wanted to pick the single speech representation that performed best.

We acknowledge that multiple representations are likely to be reflected in the MEG signal, not just the spectrotemporal modulation feature space. This is also acknowledged in the manuscript.

However, examining how prior knowledge influences multiple feature representations requires individual contributions from each feature space to be identified. Because different feature spaces tend to be correlated, this analysis is a challenging one to perform (e.g. because the variance that can be attributed uniquely to a single feature space tends to be small). We therefore have decided to focus on the feature space that explains the most variance (spectrotemporal modulation model).

As the Editors/Reviewers point out, our approach has the advantage of simplifying the analysis and interpretation of what is arguably already a complex study. We also maintain that the spectrotemporal modulation framework is ideally suited to characterising the acoustic consequences of noise-vocoding and therefore also the neural consequences.

While apriori it might be hypothesised that prior knowledge of speech content modulates speech processing at a higher (non-acoustic level) such as phonetic features (di Liberto et al., 2018a), this would be inconsistent with our previous MEG studies (e.g. Sohoglu et al., 2012; Sohoglu et al., 2016) as well as from ECoG data from other groups (Holdgraff et al., 2016; Leonard et al., 2016 *Nature Comms*). The very early latency at which we observe the interaction effect in our decoding analysis is also consistent with the notion that prior knowledge modulates processing at an early acoustic stage (Figure 6D).